# Flaxseed Lignans as Important Dietary Polyphenols for Cancer Prevention and Treatment: Chemistry, Pharmacokinetics, and Molecular Targets

**DOI:** 10.3390/ph12020068

**Published:** 2019-05-05

**Authors:** S. Franklyn De Silva, Jane Alcorn

**Affiliations:** Drug Discovery & Development Research Group, College of Pharmacy and Nutrition, 104 Clinic Place, Health Sciences Building, University of Saskatchewan, Saskatoon, SK S7N 2Z4, Canada

**Keywords:** flaxseed lignans, dietary polyphenols, phytochemicals, cellular/molecular targets, pharmacokinetics, chemopreventive, chemotherapeutic, hallmarks of cancer, quality of life

## Abstract

Cancer causes considerable morbidity and mortality across the world. Socioeconomic, environmental, and lifestyle factors contribute to the increasing cancer prevalence, bespeaking a need for effective prevention and treatment strategies. Phytochemicals like plant polyphenols are generally considered to have anticancer, anti-inflammatory, antiviral, antimicrobial, and immunomodulatory effects, which explain their promotion for human health. The past several decades have contributed to a growing evidence base in the literature that demonstrate ability of polyphenols to modulate multiple targets of carcinogenesis linking models of cancer characteristics (i.e., hallmarks and nutraceutical-based targeting of cancer) via direct or indirect interaction or modulation of cellular and molecular targets. This evidence is particularly relevant for the lignans, an ubiquitous, important class of dietary polyphenols present in high levels in food sources such as flaxseed. Literature evidence on lignans suggests potential benefit in cancer prevention and treatment. This review summarizes the relevant chemical and pharmacokinetic properties of dietary polyphenols and specifically focuses on the biological targets of flaxseed lignans. The consolidation of the considerable body of data on the diverse targets of the lignans will aid continued research into their potential for use in combination with other cancer chemotherapies, utilizing flaxseed lignan-enriched natural products.

## 1. Introduction

The exploration of alternative strategies for cancer prevention and treatment has become necessary owing to the high costs of current chemotherapies, prolonged time in regulatory authorization processes for new cancer treatments, and the considerable expenditure associated with taking a medicinal agent from bench to bed-side. Repositioning of noncancer therapeutics, such as plant polyphenols, to treat cancer offers an alternate strategy to address these challenges. Such therapeutic interventions are usually associated with lower costs and manageable toxicity with concomitant improvement in quality of life. Plant polyphenols have a long history of proposed benefits in the prevention and treatment of a chronic disease like cancer [1,2,3,4]. Although the evidence for health benefits of plant polyphenols is available throughout the literature, the flaxseed polyphenols have gained increasing attention. Flaxseed contains numerous nutrient and non-nutrient chemical constituents, like α-linolenic acid, fiber, and lignans, which can support our well-being [5,6,7,8,9,10,11,12,13,14,15,16,17,18,19,20]. More recently, polyphenols of flaxseed—the lignans—have sparked increased interest mostly attributing to their antioxidative, anti-inflammatory, anti-atherosclerogenic, and antiestrogenic potential, thus suggesting ability to reduce risk and protect against cancer [16,21,22,23,24,25,26,27,28,29,30,31,32,33,34,35,36,37,38]. Such attributes have compelled expansion of investigations into lignan mechanisms of action. 

Plant lignans [39], different from lignins (racemic polymers that are components of the plant cell wall [39,40]), are non-nutrient, noncaloric, bioactive phenolic plant compounds [39]. Diverse lignanoid constituents from plant-based food resources [41], like secoisolariciresinol diglucoside (SDG) [42], lariciresinol [41], isolariciresinol [43], 7-hydroxymatairesinol [44], matairesinol (MAT) [45], pinoresinol, arctigenin, syringaresinol [46], and asarinin [47], can be precursors of enterolignans—the mammalian-derived lignans—following oral consumption of plant lignans [39]. The primary intent of this review is to consolidate the evidence of lignan pharmacokinetics and modulation of cellular processes and cell signaling pathways within the cancer phenotype so as to provide opportunity to direct future investigations into the role and benefit of the dietary polyphenols, specifically flaxseed lignans, in the prevention and treatment of cancer (i.e., complementary and integrative medicine [48]). 

## 2. Growing Use of Naturally Derived Products

Unsatisfactory results of Western medicine have given complementary and alternative treatment options more attention [49]. Many patients rely on phytochemicals and herbal medicines (collectively referred to as natural health products (NHPs) for the purpose of this review) for primary health care, especially in the developing world [50]. In developed countries, NHPs are used to promote healthier living [50]. Although several NHPs have promising effects with wide utilization, some remain untested with clinical use unmonitored and undocumented [50], while some NHPs have safety concerns [51]. Existence of a regulatory framework for NHPs provides greater reassurance to consumers; however, regulations and product quality specification vary among countries [49]. As an example, the Dietary Supplement Health and Education Act (DSHEA) of 1994 provided the U.S. FDA the authority to implement Good Manufacturing Practices (GMP) for dietary supplements and ensure safety of such products, and the framework of the Federal Food, Drug, and Cosmetic Act, which led to the DSHEA, provides the necessary framework needed by the Food and Drug Administration (FDA) to regulate dietary supplements [52]. Additionally, in Canada, NHPs approved by Health Canada (e.g., herbal remedies [53]) are regulated under the Natural and Non-Prescription Health Products Directorate [54]. This allows for large production and lower prices (due to competition) by companies, even though NHPs are regulated somewhat similar to pharmaceutical drugs under the Natural Health Products Regulation (NHPR). These regulations protect Canadians by ensuring that the products obtained meet their health needs [55,56]. Regardless of the preclinical evidence of NHPs, translational capabilities into the clinic can be hampered by similar factors encountered by drugs in development such as the dose size and dosage forms and the variability in outcomes caused by gender, ethnicity, and comorbidities [55]. 

## 3. Cancer, the Unmet Medical Need

Cancer was identified as an important human disease thousands of years ago [57]. In the subsequent thousands of years cancer patients faced little hope for cure and survival, a situation unchanged for some cancers and clearly an unmet medical need for these patients. Globally, cancer contributes to considerable mortality with estimates projected to increase from 14 million new cases per year in 2012 [2] to an estimated 19.3 million cases yearly by 2025 [58,59]. Colorectal, liver, breast, gastric, prostate, cervical, and lung cancers remain the principal causes of cancer deaths [2], and the majority of cancer related mortality occurs in low- and middle-income countries [60,61]. Surgery, radiotherapy, and systemic therapy, which include general chemotherapy, hormonal therapy, immunotherapy, and targeted therapies, are the current treatments of cancer [58]. In too many patients these treatments fail, and cancer remains a major challenge to clinical interventions. A need exists to discover more effective ways of targeting cancer and new avenues of disease management might offer some potential. 

Today’s improved understanding of the characteristics of cancer offer renewed hope for treating cancer. The pioneering work of Weinberg and colleagues to categorize the cancer characteristics into distinct tumor properties, the so-called “Hallmarks of Cancer”, provide a framework around which to rally the considerable scientific and technological advances to identify more effective treatments for different cancer phenotypes [62,63]. Among the different cancer characteristics, mutations enable malignant cells both continuity and survival. These driver mutations stem from various mechanisms including carcinogen exposure [57]. Carcinogenesis is a complex multifactorial and multistep process separated into three closely related stages: initiation, promotion, and progression [64,65,66,67] (Figure 1). The first stage, initiation, follows usually from carcinogen (or its metabolite) exposure and is traditionally considered an irreversible step with one or more genetic alterations resulting from DNA mutations, transitions, transversions, or deletions [65,68]. Promotion is the second stage and is considered a reversible stage where the proliferation of neoplastic cells takes prominence [65]. This stage does not involve DNA structural changes but rather changes in genome expressions brought out by promoter–receptor interactions [68]. Tumor progression is the last stage where neoplastic transformation occurs followed by tumor growth, invasion, and metastasis [65]. Stages of carcinogenesis are governed by proto-oncogenes, cellular oncogenes, and tumor suppressor genes. These genes and their protein products may serve as druggable targets for cancer treatment.

In order to reduce the incidence and mortality of different malignancies, effective preventive strategies that impede tumorigenesis are needed [69]. Although the heterogeneity of tumors and tumor development may pose a challenge for successful therapeutic interventions [70], the initial stages of carcinogenesis is usually associated with a lower burden of molecular and cellular aberrations such that chemopreventive or early chemotherapy is more likely to achieve therapeutic efficacy as compared with treatment of more advanced stages of tumorigenesis [70]. Furthermore, it is well known that cancer development and progression is associated with inflammation. Hence, early preventive or treatment strategies should include anti-inflammatory therapies. While recruitment and activation of inflammatory cells due to mutations that initiate cancer may trigger cancer-intrinsic inflammation, a multitude of factors (e.g., toxin exposure, microbial infections, autoimmune disease, and obesity) are responsible for cancer-extrinsic inflammation [71]. Epidemiologic studies reveal that ~20% of all cancers emerge as a direct result of long-standing inflammatory disease [71,72,73]. For these reasons inflammation is a frequent mechanism of action for diverse cancer risk factors [71]. Therefore, various anti-inflammatory agents such as selective cyclooxygenase-2 inhibitors, nonsteroidal anti-inflammatory drugs, and natural health products with anti-inflammatory properties have been identified as potential chemopreventive agents [69,71,74,75,76,77,78,79]. 

## 4. Cancer Prevention 

The sequence of events in the multistage process of carcinogenesis provides opportunities for intervention with the goal of preventing, reversing, or delaying tumor development and progression [80]. Interventions generally fall into three categories of prevention, namely primary (preventing disease or injury), secondary (reducing impact of disease or injury), and tertiary (reducing impact of ongoing disease or injury having lasting consequences) [81,82,83,84,85,86,87,88]. These categories are based on the concept of chemoprevention first proposed in the early 1970s by Sporn [85,89], and extended by Wattenberg, who suggested the selective inhibition of carcinogenesis in any of the phases of cancer—initiation, promotion, or progression [90,91]. Primary chemopreventives block the disease by inhibiting mutagenesis, cancer initiation, and tumor promotion [65]. During early stages of tumorigenesis, secondary chemopreventive agents inhibit tumor progression by interfering with signal transduction, hormones, angiogenesis, antioxidant activity, and immune status [65]. The third class promotes chemoprevention by blocking cancer invasion and metastasis in patients usually after initial therapy [65] through mechanisms including activation of antimetastatic processors and modulation of cell adhesion factors or extracellular matrix degradation components [3,65,88] (Figure 1). However, interventions that interfere with all three phases will likely bring about a more meaningful degree of cancer prevention [81].

In general, chemoprevention can be achieved through reduction in bioactivation of procarcinogens, obstruction of expansion of additional malignant cells, or through suppression of metabolism of specific compounds to reduce toxicity [4,92]. This understanding has led to four notable categories of chemoprevention and include medications, hormones (i.e., antiestrogens and antiandrogens), vaccines, and dietary agents [93]. Only a handful of agents have been clinically approved for cancer chemoprevention (e.g., the anti-inflammatory drugs, aspirin, celecoxib and diclofenac) [70,71], with several others suggested as possible chemopreventive agents (e.g., the anti-hypercholesterolemic statins, the antidiabetic drug metformin, and antiosteoporosis bisphosphonates) [94]. The complexity associated with cancer pathogenesis has otherwise limited our ability to identify primary, secondary, or tertiary interventions that effectively reduce cancer risk or progression. The cost of patient survival and quality of life, though, continue to drive research into effective chemopreventive interventions [70,71,91,95]. 

An active avenue of research in chemoprevention involves natural chemicals derived from plants (i.e., phytochemicals). Over 80,000 species of plants are utilized in healthcare management, while more than 60% of the existing anticancer drugs come from nature [96]. The broad selection of biologically active, structurally different natural compounds continues to aid the process of cancer drug discovery with respect to chemoprevention and chemotherapy [97,98]. An abundance of phytochemical constituents with preventive anticancer properties against cancers such as lung, breast, ovarian, prostate, thyroid, and colon have been reported throughout the literature [92,99,100,101,102,103,104,105,106,107,108,109]. These phytochemicals have not seen wide application despite the limitations of current treatment methods [110] as general Western practices often dismiss their value for patient treatment. For this reason, researchers covering a wide area of health research have turned their focus on alternate ways to address the issues related to general western practices and to capitalize on the protective effects of phytochemicals [111].

## 5. Alternate Approaches to Malignant Disease

The Halifax Project—an international task force comprising of 180 scientists—has posed a “broad-spectrum therapeutic approach” as an alternate low-toxicity strategy to mitigate the problems of cancer chemotherapy [58]. Following from a rigorous examination of the cancer hallmarks, this interdisciplinary group identified 74 high-priority targets. Many of the suggested therapeutic approaches for these targets were phytochemicals with evidence of low toxicity [58]. Such phytochemicals are also commonly considered complementary and alternative medicines (CAMs), and are associated with integrative medicine. For cancer, integrative medicine is based on a foundation of lifestyle therapies, drawing attention to diet, dietary components, and physical activity [58,112,113]. It focuses on patient quality of life and demands marshalling of all therapeutic and lifestyle strategies to ensure the best outcomes and optimal health of the patient [112,114]. Phytochemicals as CAMs should be included as a strategic lifestyle intervention in a broad-spectrum approach to cancer disease management [58]. Already, the potential psychological and socioeconomic benefits of CAM use is exemplified by studies that report 32–66% of cancer patients having used cost-effective CAMs as a means to improve quality of life and therapeutic outcomes [115]. Furthermore, studies employing a combination of clinically relevant chemotherapeutic drugs with natural bioactive compounds demonstrate enhancement in antitumor effects and reduction in side-effects [111,112,116]. Some reports also document the potential of phytochemicals in overcoming chemoresistance and radioresistance of malignant cells [111,117]. Hence, the repositioning of traditionally considered “noncancer”, nontoxic phytochemical therapies with promising antineoplastic characteristics may help achieve better therapeutic outcomes and reduced toxicity profiles [118]. Convincing practitioners of this broad-spectrum therapeutic approach will be important to ensure a larger number of patients achieve improved quality of life and cancer treatment outcomes with phytochemical interventions. 

## 6. Potential of Dietary Phytochemicals for Malignant Disease

The World Health Organization (WHO) reported that approximately 65% of the world’s population relied on plant-derived drugs for their primary health care by 1985 [119]. These therapies demonstrate potential, but their safe and rational use in Western medicine is limited by a lack of rigorous scientific investigation of their potential therapeutic and adverse effects, mechanisms of action, and interactions with pharmaceuticals and functional foods [50]. Although the use of dietary phytochemicals in cancer treatment has had a long history, their efficacy is variable due to their complexity, their poorly defined targets and modes of action, and lack of knowledge of effective doses [120]. Nonetheless, a number of phytochemicals have been applied successfully in the clinical setting such as metformin and nonsteroidal anti-inflammatory drugs (NSAIDs) [121,122]. As well, the complexity and diversity in structure of phytochemicals make these compounds an often exploited scaffold to aid the discovery and synthesis of analogs that share similar structures but with improved and modified efficacy [123,124,125]. High-throughput screening (HTS), a specialized tool using automation to screen compound libraries against the drug target within a short period of time [126], has made the rediscovery of phytochemicals even more feasible [55]. The pleiotropic, multitarget effects of phytochemicals as well as polypharmacology also resonate within the emerging paradigm in drug discovery [127,128,129,130]. These factors identify dietary phytochemicals as an invaluable resource for new treatment options in current unmet medical needs, such as cancer. Yet few randomized clinical trials document the use of dietary phytochemicals in combination with standard of care treatments against human malignancies. An ability of phytochemicals to enhance the efficacy of standard treatments against cancer warrants an investigation into the wide range of biologically active compounds that have been isolated, identified, and tested for their application as treatments for cancer [131]. 

### 6.1. Dietary Polyphenols as Principal Phytochemicals for Malignant Disease

An inverse relationship exists between the high consumption of fruits and vegetables and a reduced risk of cancer [132,133], with an average 35% of all human cancer mortality directly attributed to diet [4,134]. Such statistics prompted organizations such as the WHO, the American Cancer Society (ACS), the American Institute of Cancer Research (AICR), and the U.S. National Cancer Institute (NCI) to establish dietary guidelines in an attempt to reduce cancer risk [4]. These guidelines are complemented by ongoing clinical trials that investigate diet and dietary supplements for the prevention of cancer [4]. Although food is generally perceived as providing nutritional value, phytochemicals have an additional potential to modulate molecular and cellular targets [135]. Their influence on biological function suggests that institution of an adequate, economical, and rapid system for evaluation and testing of phytochemicals with potential anticancer properties may augment the current dietary guidelines or identify lead compounds for drug discovery in different cancer phenotypes [96,136].

Phytochemicals (“phyto” in Greek means plant) are bioactive non-nutritive chemical components of plant-based diets such as fruits, vegetables, nuts, and grains [4,137] produced as primary and secondary metabolites of the plant [65]. These are generally classified into polyphenols, alkaloids, carotenoids, and organosulfur compounds [135,138] (Table 1). Primary metabolites are involved in plant functions such as respiration, development and photosynthesis, while secondary metabolites play a role in defense against herbivores and pathogens, attracting pollinators, and protection against ultraviolet radiation [139]. These secondary metabolites can have benefit in vertebrates as chemopreventive agents, drugs, herbicides, and antibiotics [65,139], and their chronic exposure is suggested to have health benefits for neurodegenerative disorders, cancer, diabetes and cardiovascular disease [140,141,142]. Polyphenols are an important class of beneficial secondary metabolites found in food and drink sources from vegetables, fruits, nuts, spices, grains, coffee tea, and wine [65]. 

Plant polyphenolic secondary metabolites are synthesized from carbohydrates through the shikimate pathway [143,144]. Although these metabolites may exist as insoluble or bound forms [144], they are present generally as glycosylated forms with single or multiple sugar or carbohydrate residues conjugated to a hydroxyl functional (–OH) group or an aromatic ring involving a co-translational or post-translational enzymatic process. Over 8000 plant-based polyphenols have been identified [65,145], and are divided into a number of classes based on chemical structure, source, and biological function including the flavonoids (flavonols, flavones, flavanones, catechins, anthocyanidins, and isoflavones), phenolic acids (benzoic acids and cinnamic acids), stilbenes, lignans, coumarins, tannins, and other polyphenols (e.g., curcumin, rosmarinic acid, gingerol) [137,139,141,146]. More broadly, polyphenolics can be classified as either being flavonoid and nonflavonoids based on their abundance [65,139]. There are over 4000 types of diverse flavonoids accounting for ~60% of structurally-related dietary polyphenols [80,141], while ~30% of dietary polyphenols are phenolic acids (i.e., hydroxy-cinnamic and hydroxy-benzoic acids) [80,141]. Flavonoids, phenolic acids, stilbenes and lignans are the most abundantly occurring plant polyphenols [80].

The literature provides ample evidence for the anticancer properties of polyphenols [2,140,147,148,149,150]. The key anticancer characteristics of polyphenols include anti-inflammatory and antioxidative effects, immunomodulation, and modulation of molecular/cellular targets within signaling pathways involved with cell proliferation, survival, differentiation, angiogenesis, migration, and hormonal activities [2,151,152]. In general, the pleiotropic effects of dietary polyphenols usually follow from their multitarget effects having the ability to impact an entire process or several processors of the malignant disease condition or status. Polyphenols typically exhibit low to moderate affinity for their targets. However, their ability to simultaneously modulate multiple targets with low affinity is suggested to account for their effects in the cancer phenotype [153,154]. Since bioactivity is not only dependent upon the interaction of the polyphenol with its target sites, but also on the chronic exposure to the polyphenol, the increasing popularity of polyphenols have led to the emergence of two new terms, ‘nutridynamics’ and ‘nutrikinetics’ [155,156]. These terms, similar in meaning to drug pharmacodynamics and pharmacokinetics, are expected to make significant contributions in our understanding of the relationship between disease phenotypes and bioactivity, as well as the interplay between chronic exposure and the host’s physiology including digestion, metabolism, and gastrointestinal microflora [157,158]. 

### 6.2. General Properties of Polyphenols and Evidence on Health 

As drug discovery efforts continue to move away from single target drugs, the multitarget characteristics of polyphenols, such as the lignans, warrant further attention to fully grasp their potential use in the clinic. Diet-derived polyphenols have gained popularity among nutritionists, food scientists, and consumers during recent years for their health-promoting and chemopreventive properties [141,159]. The beneficial effects on human health by long-term polyphenol rich diet consumption is linked to the modulation of cell proliferation, body weight, chronic disease, and metabolism [160]. The antioxidant and anti-inflammatory potential of polyphenols as indicated in animal, human, and epidemiologic studies, suggest chemopreventive or therapeutic effects for a number of noncommunicable diseases such as neurodegenerative disorders, obesity, diabetes, cardiovascular disease, osteoporosis, gastrointestinal issues, pancreatitis, and cancer [160,161,162]. Overconsumption of dietary polyphenols, especially when they are not consumed in a form of a food matrix, though, may result in adverse effects on health [160,163,164]. Our understanding of the mechanisms underlying the potential health benefits largely arise from *in vitro* studies and, therefore, a certain degree of uncertainty exists if these mechanisms hold true in human patients [160,165,166,167,168]. Nonetheless, polyphenol mechanism of action has greater complexity than the long standing belief that polyphenols form stabilized chemical complexes to negate free radicals and prevent further reactions [160,169], or result in the production of hydrogen peroxide (H_2_O_2_) for protection against oxidative stress to aid in the immune response and modulate cell growth [160,169,170]. 

#### 6.2.1. General Pharmacodynamic (or Nutridynamic) Effects of Polyphenols

In general, nutridynamic effects of polyphenols can be broadly summarized and grouped based on the following general molecular mechanisms [92]; (a) modulation of phase I and II drug metabolizing enzymes (e.g., cytochrome P450s and UDP-glucuronyltransferases) [69,80,141,171,172,173]; (b) inhibition of reactive oxygen species and modulation of antioxidant activity [4,141,171,174,175,176]; (c) inhibition of multidrug resistance (e.g., c-Myc and HDACs) [4,80,141,176,177]; (d) modulation of inflammation [69,141,172,175,177]; (e) modulation of androgen and estrogenic activity [141,176,178,179,180,181]; (f) inhibition of tyrosine kinases [80,141,176,177,182]; (g) modulation of matrix metalloproteinases, epithelial-to-mesenchymal transition [183], and metastases [80,91,141,172,177]; (h) modulation of angiogenesis [91,141,171,177,184]; (i) inhibition of cell cycle regulators and induction of cell cycle arrest [80,141,171,177,185]; (j) induction of apoptosis [80,91,141,171,175]; (k) inhibition of cell growth and proliferation [91,141,174,175,177]; (l) modulation of endoplasmic reticulum-stress and type II programmed cell death or autophagy [141,175,176,185,186,187]; (m) modulation of mitogen-activated protein kinases [69,141,171,176,177]; (n) modulation of PI3K-AKT signaling [4,69,141,177,185]; (o) modulation of JNK pathway [80,141,176,177,185]; (p) modulation of glucose and lipid [69,171,174,185,188,189]; and (q) hepatoprotective effects [190,191,192,193,194]. However, only a few polyphenols (e.g., flavonoids) have gained approval as NHPs, some with defined health claims, and none have been widely approved for clinical use [92].

#### 6.2.2. General Pharmacokinetic (Or Nutrikinetic) Characteristics of Polyphenols

Absorption and disposition (i.e., nutrikinetics) characteristics play an important role in exposure to dietary polyphenols and their eventual therapeutic effects. With oral consumption, nutrikinetic processes ultimately determine the concentration and persistence of polyphenolic compounds at their target sites. Since both genetic and epigenetic factors influence the nutrikinetics of polyphenols, these factors often result in considerable interindividual variation in blood and tissue exposure levels [137,195,196,197,198,199,200]. Despite the importance of nutrikinetics as a determinant of polyphenolic action, only a handful of *in vivo* studies have systematically addressed the factors that contribute to the differences in their absorption and disposition characteristics [137]. 

Dietary polyphenols must become systemically available to influence cancer treatment. Many plant polyphenols first undergo modification by gastrointestinal enzymes and/or bacteria to produce metabolites that are more or less systemically biologically active. The initial metabolic transformations typically involve deglycosylation to release aglycones into the gastrointestinal tract lumen following enzymatic breakdown of polymeric forms with subsequent deconjugation of monomeric forms by β-glucosidases on the brush border membrane or by the resident (small intestine and colon) gut bacteria [137,143,144]. These aglycones may undergo absorption or be further subjected to microbial enzymatic transformations including ring fission, α/β-oxidation, dihydroxylation, dehydrogenation, and demethylation reactions [137,144,201,202,203], with their subsequent absorption from the gastrointestinal lumen. Given their interactions with intestinal bacteria, polyphenols also can induce intestinal microbial changes [144], with reports that identify a polyphenol–gut microbiota interaction that either contributes to or prevents the development of disease [144,204,205]. 

During their permeation across the intestinal epithelium or with passage through the liver, aglycones or their metabolites may undergo extensive first-pass metabolism. These metabolic transformations typically involve conjugation reactions, with glucuronic acid or, to a lesser extent, with glutathione or sulfate [137]. UDP-glucuronosyltransferases (UGT), sulfotransferases (SULTs), and glutathione-S-transferases (GST) carry out conjugation reactions in both enterocytes and hepatocytes to produce conjugates that are excreted into the bile or become systemically available with subsequent excretion by the kidney into the urine [137]. Conjugates excreted into bile may undergo enterohepatic recycling making available the nonconjugated form for absorption following deconjugation by intestinal and/or microbial β-glucuronidase [137]. Typically, the aglycones are more biologically active, but the glycosidic forms, and rarely the glucuronide conjugates, have biological activity [137,206,207,208,209,210,211,212,213]. 

An important consideration in the oral bioavailability of phytochemicals is the role of intestinal epithelial transporters. Plasma membrane ATP-binding cassette (ABC) transporters play a vital role in the systemic availability of a number of dietary polyphenols or their metabolites. These ATP-dependent transmembrane efflux transporters are expressed on the apical or basolateral epithelial membrane, depending on the isoform. On the basolateral membrane, ABC transporters actively efflux phytochemical conjugates from intestinal cells (where conjugation occurred) into the portal blood supply. When expressed on the apical side of the epithelium, ABC transporters efflux phytochemicals back into the intestinal lumen to cause reductions in oral bioavailability [137]. P-Glycoprotein (Pgp/ABCB1/MDR1), multidrug resistance proteins (MRPs/ABCCs), and the breast cancer resistance protein (BCRP/ABCG2) are the key ABC efflux transporters [137,214,215] known to influence systemic availability of a number of dietary polyphenols [216,217]. For example, enterolactone is a substrate and competitive inhibitor of ABCG2 [218]. These ABC transporters exhibit several genetic polymorphisms that may influence the systemic availability of these compounds, which can contribute to considerable interindividual variation in their oral bioavailability [219,220,221]. 

Members of the solute carrier (SLC) family of transporters also contribute to the intestinal epithelial uptake of certain dietary polyphenolic compounds. The polar glycosidic forms of the dietary polyphenols typically exploit SLC transporters to ensure their systemic availability [142,222,223,224,225]. Glucosides could be transported into enterocytes by sodium-dependent glucose transporters (SGLT) such as SGLT-1 [142]. Once inside they can be hydrolyzed by cytosolic β-glucosidase to their aglycone forms [142]. Additionally, in the small intestine brush border membrane, extracellular hydrolysis of several glucosides can be carried out by lactase phloridzine hydrolase [142]. Although it is speculated that both enzymes are involved this process, their relative contribution towards different glucosides is unknown [142]. The aglycone metabolites have sufficient lipophilicity for passive diffusion to be the principal transport process during absorption. However, some polyphenols show ability to inhibit SLC transporters to influence the uptake of substrates of such transporters [215].

The relatively limited information on the tissue distribution of dietary polyphenols largely comes from preclinical evaluations with rodent models. Polyphenol compounds generally accumulate in highly perfused tissues such as the liver, kidney, heart, lung, and intestine, and many are present predominantly in their conjugated forms [142,145,226,227,228,229,230,231]. Tissue specific accumulation is observed as well, as is the case for flaxseed lignans which accumulate in prostate and breast tissue [232,233]. Extent of plasma protein binding, which can function to limit availability of polyphenols to tissue sites depending upon relative affinity between plasma protein and tissue binding sites, tends to increase with increasing lipophilicity of the compounds [213,230,234,235,236,237]. However, the polar conjugates of dietary polyphenols exhibit very limited plasma protein binding characteristics. Finally, expression of ABC transporters at tissue–blood barriers might limit access of certain polyphenols to such tissues preventing their accumulation and possible activity at such sites. For example, tumor cells typically overexpress ABC transporters to restrict access of a broad range of chemically unrelated pharmacological therapeutics to the cancer cell (aka multidrug resistance or MDR) [215]. ABC transporters also function to reduce intracellular concentrations of polyphenols, but as competitive inhibitors [215] polyphenols may enhance the cellular concentration and pharmacological response of chemotherapeutic drugs [238,239,240].

Most polyphenols are eliminated by intestinal and hepatic metabolism [241,242]. Aglycones of plant polyphenols that bypass first-pass metabolism are typically eliminated via hepatic phase II metabolism with the subsequent excretion of these metabolites by the biliary or renal system [241,242,243]. First-pass and systemic phase II metabolism are typically considered inactivation processes that result in loss of biological activity [241]. Very limited phase I metabolism occurs yielding primarily aromatic hydroxylated metabolites largely mediated by the cytochrome P450 enzyme superfamily [241,244,245]. These hydroxylated metabolites undergo further phase II conjugation and subsequent excretion by the kidney [241,242]. Given the extensive first-pass metabolism and ability to undergo enterohepatic recirculation, fecal excretion represents a major route of elimination of many dietary polyphenols, while fecal and urinary excretion is the principal route of elimination for the metabolites of polyphenols [231,241,246].

## 7. Challenges Associated with Cancer Prevention and Dietary Polyphenols

Screening, early detection, and chemoprevention are widely accepted as the major strategies to address the burden of cancer [85]. Chemoprevention as a major strategy is viewed less optimistically, as there exists a lack of clear understanding of the benefits of chemoprevention. The clustering of various approaches including dietary manipulation, NHPs such as polyphenols, and repurposed “benign drugs” into the idea of chemoprevention has done little to mitigate the uncertainty associated with the value of chemoprevention in reducing cancer risks [85]. Nonetheless, preventive strategies support the rationale for the early disruption of the carcinogenic process, which can avoid the treatment difficulties arising with the complexity and heterogeneity of more advanced stage cancers [85]. Chemoprevention has potential to address the many intrinsic genetic, epigenetic, and environmental factors that influence individual risk for cancer [85,247,248,249,250,251,252]. Hence, a basic rethinking of the nature of carcinogenesis as proceeding in a nonlinear fashion with irregular interruptions owing to changes in genetics and epigenetics [85] may benefit and expand the understanding and application of chemoprevention [85]. 

A variety of obstacles have hindered the development of dietary polyphenols as clinically approved chemopreventives. There has been a general inability to confirm their effect in reducing cancer risk due to a number of factors that include the lack of appropriate experimental models, the costs and time associated with epidemiological studies, as well as variations in length of exposure and adherence data in clinical populations, difficulties in evaluating the dietary intake of polyphenols, the variation in composition of polyphenols among different dietary sources, degradation or alteration of polyphenol chemical structures that may result in loss of bioactivity, variability and unpredictability in pharmacokinetic profiles, the impact of the microbiome on polyphenol bioactivity, and drug–polyphenol interactions [92,160,230,253,254,255,256,257,258,259,260]. To address these obstacles, research is focusing on improved extraction and purification methodology [92,261,262,263,264], development of microbial production systems for plant polyphenols [265,266,267,268,269,270], formulation of polyphenols into micro- or nanodelivery systems [271,272,273,274], development of antibody directed enzyme prodrug therapy approaches [275,276,277,278], administration of glycosidic derivatives [92,279,280], use of bioenhancers [281,282,283] or specific ATP binding cassette transporter inhibitors [92,284], use of antibiotics or other natural products to modulate intestinal microflora [260,285], development of novel polyphenol derivatives by modification of chemical structures [286,287,288,289,290], and polyphenol complexing with protein or phospholipids [92,291,292,293,294,295]. To date though, much of the investigation into polyphenols have involved *in vitro* evaluations and a sparsity of clinical trials using well-defined amounts of polyphenolic compounds [296] suggests a need for well-conducted clinical investigations to resolve their safety and efficacy as chemopreventives. The substantive evidence that exist beyond their publicized antioxidant properties [259,297], for their selective actions on a plethora of cellular and molecular signaling pathways, warrant their investigation in human clinical populations so that we may realize the health benefits of polyphenols in chronic diseases such as cancer [298,299,300,301]. 

## 8. Polyphenols of Flaxseed as Important Phytochemicals in Malignant Disease

Flax is well known for its usefulness as a source for industrial oil and fiber [109,302]. Canada and the United States are among the top producers of flax [15]. Flaxseed is considered to be a multicomponent system consisting of plant-based dietary fiber (insoluble 20–30% and soluble fiber 9–10%), oil (triacylglyceride fatty acid typically include linolenic 52%, linoleic 17%, oleic 20%, palmitic 6%, and stearic 4% acids), minor lipids and lipid-soluble components (tocopherols, monoacylglycerides, diacylglycerides, sterols, sterol-esters, phospholipids, waxes, free fatty acids, and carotenoids), protein, soluble polysaccharides, vitamins, minerals, lignans, and other phenolic compounds [15,303,304]. Various flaxseed products, such as whole and ground flaxseed, defatted flaxseed meal, and flaxseed oil, are available with suggested health benefits [305,306,307,308,309,310,311,312,313,314] and even health claims [308,315,316,317,318]. Although these products contain a number of bioactive substances including α-linolenic acid and the linusorbs (cyclolinopeptides) [5,319,320,321], lignans receive increasing attention for their health effects [322]. 

The lignans of flaxseed were once marketed in a highly concentrated standardized formulation as BeneFlax^®^, a ∼38% secoisolariciresinol diglucoside (SDG)-enriched product (Archer Daniel’s Midland) approved by both the U.S. Food and Drug Administration Agency and Health Canada that ensured a significant source of lignan with oral consumption [323]. BeneFlax^®^ demonstrated good tolerability and safety with long-term supplementation [323,324]. Additionally, Goyal et al. exhaustively listed the traditional and medicinal uses of different flax forms such as flaxseed tea, flaxseed flour and flaxseed drink for various health conditions as well as of various medicinal preparations using flaxseed oil [15]. Flaxseed oil, whole seed, ground whole seed, fully defatted flaxseed meal, partially defatted flaxseed meal, flaxseed hulls, flaxseed mucilage extracts, flaxseed oleosomes, and flaxseed alcohol extracts are among the many different types of available flaxseed products for consumption [304]. However, most of these commercially available products contain insufficient amounts of lignan, with levels of consumption unlikely to achieve therapeutic concentrations. The use of such products containing relatively modest to low lignan levels in studies investigating the health effects of lignans have contributed to inconclusive and unsatisfactory results of flaxseed lignan interventions [325,326,327,328,329]. The mounting preclinical and clinical evidence, though, suggests a need to revisit the requirement for lignan enriched products, particularly as the role of flaxseed lignans against chronic disease such as cancer continues to attract increased attention [22,33,303,330,331,332,333,334,335,336,337,338,339,340,341,342,343,344,345,346,347]. These include detailed investigations into the molecular mechanisms in order to relate to lignan safety and efficacy in malignant disease. 

### 8.1. Lignans of Flaxseed

Naturally occurring plant lignans are present in vegetables, fruits, and whole grains [22], although the major source is flaxseed (the richest known source with 9–30 mg per gram, with lignan production at 75–800 times over other sources [28,348]) followed by sesame and rye bran [349]. The seed of *Linum usitatissimum* (Linum, a Celtic word for lin or “thread,” and usitatissimum, a Latin word for “most useful” [302]) contains a rich source of the plant lignan, secoisolariciresinol diglucoside (SDG) [350], and contains minor amounts of other lignans [351,352] and cyanide-containing substances [213,303,353]. Biosynthesis of lignans in flaxseed is reported to occur through the following pathways that include phenylpropanoid pathway, stereospecific coupling by dirigent proteins, biosynthesis of dibenzylbutane lignans, and glycosylation of lignans into SDG [6,354,355,356]. SDG primarily exists in the seed hulls [350] with an average of 32 nmol/mg hull compared with 9.2 nmol/mg in the other parts of the seed [357]. Flax is an old agronomic crop with over 300 species [318]. Newer cultivars, though, can contain higher concentrations of lignans in the hull [358,359]. Differences in climatic conditions and methods of cultivation can influence the percentage composition of the various bioactive compounds [213,360,361]. In 1956, Bakke and Klosterman were first to isolate SDG from flaxseed [213]. However, scientific interest grew in the early 1970s with the discovery of enterodiol and enterolactone, later referred to as the mammalian lignans [362,363] when Axelson et al. identified SDG as the precursor for mammalian lignans [213,364]. Traditionally, plant lignans are considered phytoestrogens along with stilbenes and flavonoids containing the dibenzyl butane scaffold [213,365,366]. Today, lignan interest as potential bioactive compounds goes beyond the long-held belief of estrogenic effects as their health benefits undergo further scrutiny at the molecular level.

### 8.2. Chemistry and Pharmacokinetics (or Nutrikinetics) of Lignans

Lignans are a complex class of polyphenolic bioactive phytochemicals. Lignans can be described as stereospecific dimers of monolignols (aka cinnamic alcohols) bonded at carbon 8, which can exist free or bound to sugars in plants [39]. Secoisolariciresinol, syringaresinol, and pinoresinol are commonly found lignan diglucosides [39]. The major lignan, SDG, exists as two enantiomers—(+) and (-)—with varying distribution in different *Linum* species where some species predominatly contain one of the two enantiomers, while others contain both (e.g., *L. elegans* and *L. flavum*) [6,302]. Additionally, hydroxycinnamic acid derived monolignols can be dimerized into lignans (monolignol dimers) or polymerized into lignins (insoluble dietary fibers composed of p-coumaryl, coniferyl, and sinapyl hydroxycinnamic alcohol large polymers [367,368]) [39]. Although lignans are not categorized as dietary fibers, lignans and lignins share some chemical characteristics [39,369]. Mataresinol (MAT), SDG, lariciresinol, and pinoresinol are the most common plant lignans, but the lignans arctigenin, syringaresinol, cyclolariciresinol (isolariciresinol), medioresinol, 7′-hydroxymatairesinol, and 7-hydroxysecoisolariciresinol, sesamin, and the lignan precursor sesamolin, can also exist in plant-based food [39,138,370].

Flaxseed lignans are formed by the coupling of two coniferyl alcohol residues that become integrated into an oligomeric polymeric structure, termed the lignan macromolecule [371,373,374] (Appendix A). This polymer complex is composed of five SDG structures held together by four hydroxy-methylglutaric acid (HMGA) residues (3-hydroxy-3-methyl glutaric acid [371]) with the hydroxycinnamic acids, p-coumaric glucoside (4-O-β-d-glucopyranosyl-p-coumaric acid or linocinnamarin [371]), and ferulic acid glucoside (4-O-β-d-glucopyranosyl ferulic acid [371]) as end units linked to the glucosyl moiety of SDG [354,372,374,375,376]. HMGA is considered the backbone of the lignan macromolecule [374] (Figure 2). The flavonoid herbacetin diglucoside (HDG) is also part of the lignan macromolecule attached via ester linkages with HMGA, similar to SDG [374,377]. This complex is a straight chain oligomeric complex with an average molecular weight of 4000 Da [371] and the average chain length of the complex is three SDG moieties with a hydroxycinnamic acid at each of the terminal positions [372]. Additionally, caffeic acid glucoside (CaAG) has also been isolated from the flaxseed lignan macromolecule [378]. The suggestion that the different phenolic compounds of flaxseed exist in acylated forms adds further complexity to the lignan polymer composition of flaxseed [371]. 

The bioactivity of the lignans requires their removal from the oligomeric macromolecule structure upon oral consumption of the seed hull. The mechanism of release of SDG from the complex is uncertain, but the cleavage of the glucose groups of SDG is thought to be mediated by β-glucosidase and bacterial fermentation in the gastrointestinal tract to yield its aglycone, secoisolarisiresinol (SECO) [379]. SDG-deglycosolating bacteria strains *Clostridium* sp. SDG-Mt85-3Db (DQ100445), *Bacteroides ovatus* SDG-Mt85-3Cy (DQ100446), *Bacteroides fragilis* SDG-Mt85-4C (DQ100447), and *B. fragilis* SDG-Mt85-5B (DQ100448) are mainly responsible for conversion in the human gastrointestinal tract [55,380]. Alternative bacterial species (e.g., *Butyrivibrio fibrosolvens, Peptostreptococcus anaerobius*, and *Fibrobacter succinogens* in cow [381] and Klebsiella [382], *Bacteroides distasonis*, *Clostridium cocleatum*, *Butyribacterium methylotrophicum*, *Eubacterium callendari*, *Eubacterium limosum*, *Ruminococcus productus*, *Peptostreptococcus productus*, *Clostridium* scindens, *Eggerthella lenta* and ED-Mt61/PYG-s6 in human [179,380,383,384] and *Ruminococcus gnavus* in goat [385], *Prevotella* spp., and *B. proteoclasticus* [381]) are also involved in various reactions with lignan conversion [380,381]. SECO may undergo further bacterial demethylation and dihydroxylation reactions to produce the mammalian lignan, enterodiol (ED), which undergoes further oxidation to enterolactone (ENL) by microbes. *Streptomyces avermitilis* MA-4680 and *Nocardia farcinica* IFM10152 bacteria have the highest hydroxylation activity for ED [386]. The microorganism P450 enzyme, Nfa45180, is reported to show the highest hydroxylation activity towards ED, especially for ortho-hydroxylation of the aromatic ring *in vivo* [386]. Other plant lignans in flaxseed, such as matairesinol (MAT), pinoresinol (pinoresinol diglucoside [387]), and lariciresinol (isolariciresinol [351]) that are found in minor amounts, are also converted into the mammalian lignans ENL and ED [46,388] (Figure 3). Hence, following consumption, SDG is released from the macromolecule (the exact location within the gastrointestinal tract is unknown) and is principally converted to the mammalian lignans in the lower intestine, either by the brush border enzymatic activity of the gut mucosa or by bacterial enzymatic activity [379,389]. Oral antimicrobial drugs are known to decrease serum concentrations of mammalian lignans highlighting the importance of gastrointestinal flora in the production of mammalian lignan [390]. A detailed composition of the flax lignan macromolecule, history of lignans and the analytical methods used to identify lignans as well as extraction, isolation and purification techniques are described in previous reviews [28,391].

Bioactivity of the lignans also requires their adequate systemic exposure following oral consumption. Systemic exposure of the lignans is generally quite low due to their limited oral bioavailability. As a polar molecule, SDG does not undergo oral absorption due to its poor permeation characteristics across the gastrointestinal mucosa [199,213,234,323,392,393]. The lipophilicity of the aglycone SECO and the mammalian lignans encourages passive diffusion across the gastrointestinal mucosa [392]. However, these lignan metabolites (Appendix A) are subject to extensive first-pass metabolism, primarily through phase II conjugation reactions, before they enter the systemic circulation resulting in their rather limited bioavailability [55,323,379,392,393,394,395,396,397,398,399]. Glucuronidation by UDG-glucuronosyltransferases (UGTs) is the principal conjugative reaction, although sulfation by sulfotransferases (STs) and, to a minor extent, methylation by catechol-O-methyltransferase also contribute to lignan metabolite metabolism [246]. Although the ST isoforms involved in lignan metabolism are unknown, animal studies suggest the UGT2B subfamily is principally responsible for the glucuronidation of lignans to mono- and diglucuronic acid conjugates [55,400]. The extent of conjugation relates to the order of lipophilicity (SDG < SECO < ED < ENL) [392], and therefore indicates that metabolism, excretion, and the ratios of each conjugate and aglycone may vary depending upon the cell and tissue type. Polar conjugates of the lignans produced in enterocytes are transported out of the cell to the portal blood supply through the activity of the multidrug resistance-like protein, MRP3 [242]. Such polar metabolites bypass the liver and are ultimately excreted by the kidney, but lignans escaping intestinal first-pass metabolism undergo additional hepatic phase II metabolism and to a limited extent, cytochrome P450 enzyme mediated metabolism [46,55,213]. Relevant lignan–drug interactions have not been identified, but lignans may reversibly inhibit several cytochrome P450 enzymes at high concentrations [213,401]. Additionally, SECO and ENL can activate the nuclear receptor pregnane X receptor (PXR), which may modulate the induction of phase I and II enzymes and, in turn, alter systemic and tissue concentrations of other substrates of these enzymes [213,402]. 

Given the propensity for hepatic phase II metabolism, SECO, ED, and ENL undergo enterohepatic recirculation [403] (Figure 4). Reabsorption of nonconjugated lignans result in fluctuations in plasma concentrations, as evidenced by secondary peaks in the oral plasma concentration-time profile [213], and prolongation of their mean residence times in the body [199,213]. Approximately 20–50% of glucuronide and sulphate conjugates are excreted into the bile [213,404], and of this amount 80% is deconjugated by β-glucuronidase of intestinal microflora in the intestinal lumen [213,405]. β-glucuronidase activity has been detected in various bacterial genera such as Bacteroides, Bifidobacterium, Eubacterium, and Ruminococcus, belonging to the prominent human intestinal microbiota [406], and specifically genes encoding β-glucuronidase have been described in *Escherichia coli*, *Lactobacillus gasseri*, *Clostridium perfringens*, *Staphylococcus aureus*, and *Thermotoga maritina* [406]. Due to the high β-glucuronidase activity, such glucuronides are more likely to be hydrolyzed back to their aglycone forms for reabsorption or their fecal excretion. As a result, physiologically relevant lignan concentrations in the range of 10 to 1000 μM are likely achievable in the colon lumen [407,408]. Enterohepatic recirculation also results in 10–35% of conjugated and unconjugated lignan excretion by the fecal route [213,409,410,411,412]. However, a considerable proportion is excreted by the kidney as glucuronide conjugates with permanent removal from the body [213,413]. Generally, a good correlation exists between plasma concentrations and urinary excretion of various lignan metabolites [213,411,414], but the relative ratio and extent of urinary excretion can vary depending upon population characteristics, e.g., postmenopausal women with or without breast cancer [199,213,415,416,417]. Additionally, small portions of enterolignans have been reported to be found in certain animal based food such as milk [39,418,419,420] and therefore can be considered as another route of excretion.

Blood and tissue levels of the lignans and their conjugative metabolites show a high degree of interindividual variability due to variation in their absorption and disposition (distribution and elimination) characteristics as well as differences in diet, microflora, gender, and age [213,421]. Extensive first-pass metabolism results in concentrations of unconjugated SECO, ED, and ENL in the lower nanomolar range [323,421,422,423,424], with concentration of the conjugated forms 250 times or more higher than the unconjugated lignans [55,213,323,379,393]. Low plasma concentrations are also due to their wide distribution throughout the body with detection in tissues such as the intestine, liver, lung, kidney, breast, heart, and brain with higher levels in liver and kidney [213,232,392,404,425,426]. In humans, plasma protein binding of flaxseed lignans is unknown but in rat plasma the unbound fraction for SECO, ED, and ENL was 33%, 7%, and 2%, respectively [234]. Plasma protein binding of the conjugated metabolites of SECO, ED, and ENL is likely very low given the polar nature of these metabolites. Additionally, an erythrocyte partitioning study indicated no accumulation of ENL in erythrocytes [234]. Despite low blood levels of ENL and the polar nature of the conjugates, ENL and its conjugates seem to concentrate in body fluids such as breast milk, intestinal fluid, prostatic and breast cyst fluid [381,420,424,427,428]. Accumulation is also most prominent with chronic administration of flaxseed lignans compared to acute administrations [213,233,429]. 

Although lignan accumulation into solid tumors is unknown, tumors commonly possess poorly formed, highly permeable vasculature that results in the accumulation of various macromolecules (e.g., plasma protein albumin) within the tumor microenvironment [430,431,432]. Several studies have suggested tumors as sites of albumin catabolism, and the presence of putative albumin-binding proteins on tumor cell surfaces [433]. Therefore, taking this into consideration, it is possible that albumin bound lignans may accumulate in the tumor environment independently and/or released upon albumin catabolism, e.g., similar to albumin conjugated drugs used for increasing intratumoral accumulation of drugs for antitumor effects [434]. Similarly, the conjugated metabolites of the aglycone and mammalian lignans may gain easy access to the tumor microenvironment due to the leakiness of the tumor vasculature.

The biological interactions of SECO, ED, ENL, and their phase II conjugates within the molecular and cellular environment remains unclear. Phase II enzyme reactions are typically considered as deactivation pathways. Hence, extensive first-pass metabolism, which results in high levels of circulating phase II conjugates traditionally considered to be inactive metabolites [400], raises questions on how lignans exhibit health benefit following oral consumption. Recent evidence, though, may suggest the conjugative metabolites of the mammalian lignans exert pharmacological activity in certain cellular contexts [435]. Furthermore, evidence exists of the ability of polyphenolic glucuronide conjugates to undergo deconjugation reactions in specific tissues such as the inflammatory sites of the tumor microenvironment due to extracellular availability of β-glucuronidase, which expresses optimal enzyme activity at low pH [142,436,437,438,439,440], as well as evidence of the ability of the vascular endothelium to deconjugate certain glucuronide conjugates [436,441]. This suggests that high circulating glucuronide conjugates might act as aglycone carriers with release of the aglycones at target sites upon deconjugation [436,442].

### 8.3. Lignans as Therapeutic Agents for Cancer

Cancer involves complex mechanistic changes in multiple interdependent and redundant cellular signaling networks that ensure initiation, survival, and promotion of carcinogenesis. This complexity results in many failures of single target therapies in clinical drug development despite the enormous investments made to advance such products to the market [443]. The effects of conventional chemotherapy, though, might be enhanced by compounds that have ability to inhibit and antagonize multiple targets within the complex array of cell signaling processes [444]. This is supported by an increased focus on multitarget agents in drug discovery and development, which has gained much needed attention in recent years [445]. The historical presence in the human diet of phytochemicals such as lignans, though, may have an advantage over synthetic compounds due to their coevolutionary exposure. Given their possible multiple therapeutic effects, we have witnessed an increased investment into the investigation of their mechanisms of actions in order to more fully understand their antitumor effects [443]. 

Flaxseed lignans have a long history of purported health benefits [25,28,38,304] (Figure 5). For cancer, flaxseed is consumed for both chemopreventive and treatment purposes [15,318,446,447]. Studies with preclinical models of cancer clearly have demonstrated therapeutic benefits of lignan rich diets with evidence of reductions in early tumorigenesis [448,449], as well as inhibition of tumor growth, angiogenesis, and progression of the disease [450,451]. Such evidence supports the putative relevance of lignans in carcinogenesis [1,15,55,174]. However, clear evidence of benefit in human clinical populations is confounded by the numerous epidemiological and population-based studies that report an unclear accounting of the daily lignan dose and, hence, uncertain lignan exposure levels [325,452,453,454,455,456,457]. The availability of standardized lignan-enriched products now provides opportunity to clearly understand daily dose exposures and ensure adequate therapeutic levels for clinical benefit. Such lignan-enriched products have demonstrated good safety and tolerability in vulnerable populations, such as frail elderly adults [323,458], as well as in other preclinical and human clinical trials [32,55,459,460,461], except during pregnancy [462,463,464,465] and lactation [420], or with products that produce high ED levels [466,467]. These lignan-enriched products can guarantee pharmacological lignan doses, which will allow us to address past inconclusive epidemiological studies of the effect of lignans (and other polyphenols) on human cancer risk and therapy [55,323,326,333,417,452,468,469,470,471,472,473,474,475,476,477,478,479,480,481]. 

### 8.4. Linking Benefits of Flaxseed with Cancer Associated Chronic Diseases 

Cancer shares a number of risk factors common to other chronic disease states [482]. This is emphasized by statistics that indicate cancer, diabetes, and cardiovascular disease (CVD) were responsible for 71% of deaths globally in 2015 [482]. Chronic diseases, including type 2 diabetes [483,484,485], and CVD risk factors, such as cholesterol level [486,487,488,489,490,491], heart rate [492,493], blood pressure [494,495,496,497,498], uric acid [499,500,501,502,503], and chronic kidney disease [504,505,506,507,508,509], as well as pulmonary disease [510], are associated consistently with the risk of cancer [482]. The abundant evidence confirming the health promoting beneficial effects of flaxseed in chronic disease can be grouped according to health benefits in (1) the cardiovascular system (e.g., platelet aggregation, atherosclerosis, hyperlipidemia, and dyslipidemia [7,9,10,17,19,24,26,27,35,485,511,512,513,514,515,516,517,518]); (2) insulin resistance, glycemic control, and obesity [9,25,27,35,519,520,521,522,523,524,525,526]; (3) inflammation [5,7,8,19,25,27,514,527,528] and oxidative stress [5,12,21,23,35,407,513,514,529,530,531,532,533,534]; (4) hepatic [28,535] and renal systems (e.g., lupus nephritis) [15,28,435,536]; (5) the immune and nervous system [7,15,28,537,538,539,540,541,542,543]; (6) the reproductive system [8,11,14,25,28,33,311,352,514,542,544,545,546]; and (7) the gut microbiome [547,548,549]. Detailed discussions on the relationship between chronic disease and flaxseed can be found in our previous review [25] and others [5,6,8,15,16,21,34,304,513]. This collective epidemiological, observational, and preclinical evidence support the idea of flaxseed lignans as qualified candidates for risk reduction and treatment of chronic disease (Appendix A) warranting additional clinical trials with known pharmacological doses to provide the evidence base to support their use clinically [21,513]. 

### 8.5. Purposing Lignans into Established Models of Cancer Characteristics

Several models have been elaborated to describe the wide range of properties and characteristics of cancer [62,550]. These models aid in understanding both the complexity of cancer pathogenesis and the various processes contributing to cancer, as well as to focus research efforts on identifying possible chemopreventive agents or therapeutics [59]. These models provide an organizing framework to explain responses to a targeted therapy, where cancers may modify their dependence on a particular hallmark, while enhancing the activity of another [551]. The “hallmarks of cancer” model established in 2000 by Hanahan and Weinberg identified the six cancer hallmarks of evading growth suppressors, resisting cell death, activating invasion and metastasis, enabling replicative immortality, sustained proliferative signaling, and inducing angiogenesis [63]. This model was subsequently updated in 2011 with further inclusion of two enabling characteristics (genomic instability and tumor-promoting inflammation that support cancer cells to acquire the hallmarks) and two emerging hallmarks (deregulation of cellular energetics and avoidance of immune destruction) [59,62]. In a model (signaling pathways and cellular processes) articulated by Vogelstein et al. in 2013 [550], tumors contain two to eight “driver gene” mutations that drive cancer growth, while the remaining “passenger” mutations do not add to the selective growth advantage [59]. Genes either contain intragenic mutations (Mut-driver genes) or epigenetic alterations (Epi-driver genes), both of which are responsible for carcinogenesis as well as a selective growth advantage. According to this model, twelve major signaling pathways drive cancer growth and include (a) cell survival: PI3K (phosphatidylinositide 3-kinase), MAPK (mitogen-activated protein kinase), RAS (rat sarcoma), STAT (signal transducers and activators of transcription), cell cycle/apoptosis, and TGFβ (transforming growth factor β); (b) cell fate: NOTCH, HH (Hedgehog), APC (Adenomatous polyposis coli), chromatin modification, and transcriptional regulation; and (c) genome maintenance: DNA damage control related pathways [59]. Finally, K.I. Block’s model (pathways of progression and contributing metabolic factors) of nutraceutical-based targeting of cancer lists nine “pathways of progression” (proliferation, apoptosis, treatment resistance, immune evasion, angiogenesis, metastasis, cell-to-cell communication, differentiation, and immortality) and six “metabolic terrain factors” (oxidation, inflammation, glycemia, blood coagulation, immunity, and stress chemistry) that influence the quality of life of all cancer patients [59,552]. Together, these models clearly demonstrate the interrelationships of different signaling network pathways and the enormous number of targets that require interrogation for cancer prevention and therapeutic management. 

Many phytochemicals are known to modulate multiple targets within these complex cancer processes [444,553,554,555,556]. In particular, flaxseed lignans may concurrently target various complex interdependent pathways involved in cancer progression and survival raising the possibility that lignans could be incorporated into a design of a broad-spectrum combination chemotherapy [557] (Figure 6). Drug discovery programs today have moved away from the single-target approach and currently consider systems biology approaches to improve pharmacological network understanding [59]. The complexities in tumor heterogeneity and in the interconnections amongst the various growth factors, cytokines, chemokines, transcription factors, and the proteome makes systems biology approaches exceedingly more relevant [558,559]. It also makes the broad-spectrum multitargeted approach to cancer highly significant [59]. In recognition of this changing paradigm to cancer discovery, we compiled the known lignan targets alongside their potential identification within the different cancer characteristics models listed above (Table 2).

### 8.6. The Multitarget Effects of Lignans in Cancer (Nutridynamics)

In the following section, the ability of the lignans to influence the cancer phenotype is broadly organized according to the cancer hallmarks [62]. Lignan modulation of a specific target is highlighted under specific areas linked to hallmarks, although any one target might have overlapping function in the different hallmarks. Examples are provided, but due to the complexity and interconnections among molecular signaling networks, flaxseed lignans are able to impact an array of targets leading to the modulation of various signaling cascades in the different stages of the malignant disease to disfavor progression. The reader is referred to the review by Teponno et al. [560] for detailed information on other lignans and neolignans.

#### 8.6.1. Antioxidant and Anti-Inflammatory Properties 

Lignans are well recognized for their antioxidant and anti-inflammatory activities, key properties that contribute to their multitarget effects [55,561]. As polyphenolic compounds, lignans can act as direct antioxidants (e.g., direct scavenging of hydroxyl radical) [562] or through indirect mechanisms such as modulation of the expression of antioxidant enzymes [563,564]. A number of *in vitro* studies have shown the lignans to be effective direct antioxidants [565]. For example, the direct antioxidant activity of SECO, ED, and EL exceeded vitamin E, a typical comparator, by approximately 4.5 to 5 times, with SDG showing similar activity to vitamin E [38,530]. ED and EL are reported to be effective inhibitors of lipid peroxidation *in vitro* [407], and in a model of lipid autoxidation, SECO showed much better antioxidant activity than SDG [566]. There is no significant difference between SECO/SDG and BHT—a food preservative known to cause liver toxicity—to prevent/delay the autoxidation process [567]. SDG and SECO are effective antioxidants (attributed to the 3-methoxy-4-hydroxyl substituents) against 1,1-diphenyl-2-picrylhydrazyl (DPPH))-initiated peroxyl radical plasmid DNA damage and phosphatidylcholine liposome lipid peroxidation [532]. In an aqueous environment, benzylic hydrogen abstraction and potential resonance stabilization of phenoxyl radicals are likely to aid in the antioxidant activity of the mammalian lignans [532]. Further details on the antioxidant properties of flaxseed lignans can be found in several other reviews [38,568,569]. Despite these direct antioxidant effects *in vitro*, it is debatable whether lignans attain adequate systemic concentrations with dietary consumption to mediate similar effects *in vivo* as lignans largely exist as conjugated metabolites.

The indirect antioxidant activity of lignans is mediated through upregulation of a number of antioxidant enzymes and phase II detoxifying enzymes. Upregulation of these enzymes is associated with the nuclear factor erythroid 2 (Nrf2)-linked pathway—a key transcriptional regulator of many antioxidative and anti-inflammatory pathways [570]. Nuclear factor-κB (NF-κB) is a transcription factor that is of importance in inflammation and plays a role in development, cell growth, cell survival, and proliferation [571]. Certain NF-κB-regulated genes play a pivotal role in controlling reactive oxygen species (ROS), but ROS also has various inhibitory/stimulatory effects in NF-κB mediated signaling [571]. Transcriptional regulation by Nrf2 is clearly associated with lignan induction of heme oxygenase-1 (HO-1) expression [562] with subsequent modulation of NF-κB mediated inflammatory and oxidative pathways [572,573]. Lignans also increase the abundance of antioxidant genes such as superoxide dismutase (SOD), catalase (CAT) and glutathione peroxidase (GPX) [564,574], and induce the expression of glutathione S-transferases (e.g., GSTM1) and NAD(P)H dehydrogenase [quinone] 1 (NQO1) [575,576]. SOD2, SOD1, NQO1, CAT, GST, and GPX are all regulated by NF-κB [571]. SDG in physiological solutions provide DNA radioprotection by scavenging active chlorine species and reducing chlorinated nucleobases [577], suggesting SDG as a promising candidate for radioprotection of normal tissue during cancer radiation therapy [578]. The molecular pathways connected to these various antioxidant activities contribute to the control of multiple cancer hallmarks such as “resisting cell death”, “genome instability and mutation”, “deregulating cellular energetics”, and others depending on the context.

The anti-inflammatory properties of lignans are well-documented and are suggested to benefit chronic inflammatory diseases such as cancer [55,576,579,580,581]. Lignans can modulate inflammation through several mechanisms including modulation of immune cell activation through interference with NF-κB pathway signaling [408]; reductions in proinflammatory cytokines, such as IL-1ß, IL-6, TNFα, HMGB1, and TGFß1, and cytokine receptors, TNFαR1 and TGFßR1 [582]; and downregulation of cyclooxygenase enzyme activity and levels [583]. Flaxseed also downregulates microRNA (miRNA) miR-150, which is integrated into immune response-mediated networks [584]. Lignan influence on the inflammatory process clearly impacts the “tumor promoting inflammation” hallmark of cancer.

#### 8.6.2. Anticarcinogenic and Antimutagenic Properties

Carcinogenesis occurs in several stages and mutagenesis supports the progression of the malignant disease. As effective antioxidants against DNA damage and lipid peroxidation [532], lignans are suggested to have chemopreventive properties in cancer, and SDG is emerging as a potential anticarcinogenic agent [344,582]. Preclinical *in vivo* studies that have demonstrated decreased incidence of tumor formation in tumor induction models and reductions in the procarcinogenic microenvironment following flaxseed lignan supplementation, offer support for lignans as chemopreventive agents [583,585,586,587,588]. Activation of p53 can induce cell cycle arrest as well as apoptosis in response to DNA damage [589]. The transcriptional activation of target genes of p53 is critical in cell fate determination after genotoxic stress [589,590]. Oxygen radical-based alterations at specific nucleotides can lead to mutations that occur when altered bases are copied by DNA polymerases (replicate the genome) [591]. ROS has been attributed to the pathogenesis of liver, lung, and prostate cancers [591]. The use of antioxidant therapy (preventive), such as with the lignans [576,581,582,592], has been suggested to slow tumorigenesis to prevent clinical presentation of cancers [591]. In cancer cell model systems, lignans can modulate the percentage of cells in the different stages of the cell cycle [593], downregulate viral oncogenes E6 and E7, upregulate tumor suppressor p53, and fail to exhibit genotoxicity in cancer cells [588]. However, depending upon the cancer type, p53 status, and lignan concentration, flaxseed lignans may have different effects on cancer prevention and treatment.

Interestingly, the various signaling pathways involved in anticarcinogenic and antimutagenic effects of lignans could be connected to lignan ability to favorably modulate lipid and glucose homeostasis [9,519,537,594,595,596]. High cholesterol, fat, and glucose levels are known to increase the risk of cancer [491,597,598,599,600,601,602]. Several studies have shown altered cholesterol metabolism and accumulation within mitochondria of malignant cells seems to favor continuous cell growth, survival, and progression [603,604,605,606]. The lignans variably influence targets within cellular energy and lipid homeostasis pathways, including the ability to reduce expression and activity of CPT 1 (carnitine palmitoyltransferase 1), as well as modulate pAMPK (5’ adenosine monophosphate-activated protein kinase), PPARα (peroxisome proliferator-activated receptor alpha), FASN (fatty acid synthase), expression and activity of SREBP1c (sterol regulatory element-binding proteins) and adipogenesis-related genes, such as leptin, adiponectin, glucose transporter 4 (GLUT-4), and PPARγ (peroxisome proliferator-activated receptor gamma) [213,607,608,609,610]. Additionally, a recent study reports upregulation of INSIG-1 (insulin-induced gene 1) and alteration in intracellular cholesterol trafficking in Caco2 colorectal adenocarcinoma cells [435]. Collectively, lignan effects on cellular energy metabolism and lipid homeostasis favorably modulate the cancer hallmarks of “deregulated cellular energetics” and “resisting cell death”.

#### 8.6.3. Anti-proliferative properties 

Lignans are known to reduce chemically-induced mammary and colon tumorigenesis [31,345,582]. In addition to their well-known antioxidative and anti-inflammatory effects, lignans are purported phytoestrogens with ability to modulate estrogen receptors and other hormonal functions [611]. Their putative role as phytoestrogens prompted extensive investigation into hormone-dependent cancers, since hormones play a vital role in their etiology influencing rate of cancer cell division, differentiation, survival, and metastasis [55,612,613]. Interestingly, lignans demonstrate weak binding properties to estrogen receptor α (ERα) and ERβ suggesting a limited potential for estrogenic and antiestrogenic activity [614]. Yet, studies suggest lignans’ ability to inhibit hormone-dependent cancer cell proliferation, cancer growth, and progression [55,467,615,616,617]. This may result from such mechanisms as lignan-mediated reduction in the expression of hormonal and growth factor receptor expression or binding affinity (e.g., ER, progesterone receptor (PR), EGFR (epidermal growth factor receptor), and IGF-1R (insulin-like growth factor 1 receptor)) [341,453,618], regulation of plasma sex hormone binding globulin (SHBG) levels [55,619] or binding affinity with endogenous hormones [55,620], competition with estradiol for the type II estradiol binding sites (EBS) [621,622,623], inhibition of aromatase and 17β-hydroxysteroid dehydrogenase and thereby reducing sex hormone synthesis [55,624,625,626], modulation of secreted matrix metalloproteinase (MMP) activities [531], and/or alteration in the expression and activity of cell cycle regulators and signal transduction networks regulating cell proliferation, survival, and migration [335,593,613,618,627,628,629]. The multitarget effects of lignans on hormonal signaling pathways identify their key role in modulating the important cancer hallmark of “sustaining proliferative signaling”. 

#### 8.6.4. Dysregulated cellular metabolism

A common feature of cancer cell metabolism is the ability to obtain nutrients from the nutrient-poor tumor environment to maintain viability and make new biomass [630]. Given the linkage between cell proliferation and cell metabolism [631], the core fluxes such as aerobic glycolysis, *de novo* lipid biosynthesis, and glutamine-dependent anaplerosis, have been suggested to form a stereotyped platform in order to carry out proliferation [631]. Additionally, regulation of these cellular fluxes are predominantly linked to phosphatidylinositol 3-kinase (PI3K)/protein kinase B (Akt)/mechanistic target of rapamycin (mTOR), hypoxia-inducible factor 1 (HIF-1), and Myc (myelocytomatosis oncogene) mediated signal transduction and gene expression [631]. Interestingly, upregulation of HO-1 is suspected to act through PI3K/Akt and Nrf-2 signaling pathways [632,633]. PI3K/Akt signaling (master regulator of glucose uptake) stimulates mRNA expression of GLUT1 glucose transporter and the translocation of its protein to the cell surface [630]. Akt amplifies the activity of the glycolytic enzymes hexokinase, the first enzyme of the glycolytic pathway (phosphorylates glucose molecules, and prevents their efflux out of the cell), and phosphofructokinase (catalyzes the main irreversible step) [630]. Akt alone also is capable of stimulating glycolysis to restore cell size, viability, mitochondrial potential, and ATP levels [630]. Additionally, constitutively active Akt can prevent reductions in ATP levels, which is usually triggered by the loss of cellular attachment [630]. Lignans have been reported to reduce Akt signaling [618,629,634], as well as HO-1 and Nrf-2 signaling [562]. Therefore, not only the hallmark of “sustaining proliferative signaling” but other hallmarks such as “deregulating cellular energetics”, “enabling replicative immortality”, and “evading growth suppressors” can be targeted by lignans.

#### 8.6.5. Antiangiogenic Properties 

Angiogenesis is a complicated process that depends on the type of tumor [635,636]. Solid tumors with high vascularization (e.g., ovarian cancer, non-small cell lung cancer, renal cell carcinoma, hepatocellular carcinoma, and colorectal cancer) have been the main focus of the development for antiangiogenic drugs [636,637]. The series of events in this complex process include an initial activation of endothelial cells (EC), which often results in the release of proteases that causes the degradation of the basement membranes in the surrounding area of existing vessels, and the migration of ECs to the growing lesion, followed by extensive cell proliferation forming tubes for new blood vessels [635]. However, unlike normal tissue angiogenesis, tumor blood vessel network is disorganized and leaky [638]. Lignans may have a role as effective agents in targeting the hallmark “inducing angiogenesis”. Lignans were shown to inhibit estradiol-induced tumor growth and angiogenesis *in vivo* [451]. The antiangiogenesis activity may relate to ability of lignans to reduce extracellular cancer stroma-derived vascular endothelial growth factor (VEGF) and increase in placenta growth factor (PIGF), a VEGF family member [331,451]. The platelet-derived growth factor (PDGF), its receptor, PDGFR, fibroblast growth factor (FGF) and its receptor, FGFR pathways, can aid in compensatory escape mechanisms facilitating tumor growth from anti-VEGF/VEGFR therapy drugs, which has been the gold standard pharmaceutical target [636]. However, current antiangiogenic strategy is investigating novel and emerging agents that target multiple pathways for treatment [636]. Interestingly, lignans also modulate PDGF signaling pathways making it a multitargeted agent to suppress tumor growth [628].

#### 8.6.6. Anti-invasive and Antimigratory Properties 

The lignans can modulate a number of key targets to reduce cancer cell propensity for invasion and migration [13,32,55,330,331,467,618,639]. Lignans were shown to reduce metastasis in an experimental model of melanoma [108,582]. They demonstrate ability to inhibit matrix metalloproteinases (MMPs), the enzymes responsible for degradation of the extracellular matrix (ECM) [55,640,641,642], modulate the phosphorylation of FAK (focal adhesion kinase), Src (proto-oncogene nonreceptor tyrosine protein kinase Src), and Paxillin, with subsequent modulation of their key targets (e.g., uPA (urokinase-type plasminogen activator), PAI-1 (plasminogen activator inhibitor-1), TIMP-1 (TIMP metallopeptidase inhibitor 1) and TIMP-2, RhoA (Ras homolog gene family, member A), Rac1 (Ras-related C3 botulinum toxin substrate 1), Cdc42 (cell division control protein 42 homolog), and ITGA2 (Integrin subunit alpha 2)) [627,628], and inhibit organization of the actin cytoskeleton to influence cell motility and clonogenicity [627,628,642,643]. Given that cancer relapse and metastasis continue to challenge effective chemotherapy [644,645], such properties suggest a potential for the use of lignans to target this cancer hallmark.

#### 8.6.7. Induction of Apoptosis and Cell Death 

Apoptosis plays a pivotal role in the pathogenesis of cancer where limited apoptosis results in survival of malignant cells. The complex mechanism of apoptosis is linked to many cell signaling pathways where deregulation can cause malignant transformation, metastasis, and resistance to anticancer drugs [646]. Consequently, lignan-mediated enhancement of apoptosis can occur through many mechanisms that are generally categorized into disruption of mitochondrial membrane potential (mitochondrial mediated cell death) [55,629,634], and activation of the intrinsic or extrinsic apoptotic pathways through mechanisms such as TRAIL (tumor necrosis factor (TNF)-related apoptosis-inducing ligand)-induced BID (BH3 interacting-domain death agonist) cleavage [629], reduction in antiapoptosis proteins, Bcl-2 (B-cell lymphoma 2) and survivin [22,588], caspase dependent cell death [634], and death receptor-sensitization through decreased expression of death receptor DR4 expression and TRAIL-DISC (death-inducing signaling complex) proteins, c-FLIPL/S (cellular FLICE-inhibitory protein: short form; FLICE: (Fas-associated death domain-like interleukin 1β-converting enzyme) and caspase-8, and pGSK-3β (glycogen synthase kinase 3 beta) [629,634]. Flaxseed along with radiation therapy have reported to significantly decrease the p53-responsive miRNA, miR-34a, which is responsible for regulating cellular senescence and apoptosis related factors [584]. Dietary flaxseed lignan complex, mainly consisting of SDG, induced radiosensitizing effects in a model of metastatic lung cancer. SDG is protective against radiation pneumonopathy, decreasing lung injury and eventual fibrosis, while improving survival indicating its ability to selectively target malignant cells but spare normal cells [576,581]. Although specific targeting of apoptosis can be associated with safety issues [646], as one of multiple hallmarks influenced by lignans, the ability to enhance cell death is an important attribute of the role of lignans in the therapeutic management of cancer. 

## 9. Final Remarks

Cancer remains a significant unmet medical need despite the extensive research into possible pharmaceutical solutions to tackle the various cancer phenotypes. Unfortunately, cancer will continue to be an important cause of morbidity and mortality in the near future as we witness increasing urbanization, increasing life expectancy, changing lifestyle, globalization, and changing environmental factors [4,654,655,656,657,658]. To address this global health dilemma, we may need to adopt a “broad-spectrum therapeutic approach” into our chemopreventive and therapeutic plans of cancer mitigation. Such a trend is already being observed as the 2012 U.S. National Health Interview Survey (NHIS) reported over 30% of adults and 12% of children used atypical approaches to health care [48]. The application of plant-derived bioactives or phytochemicals for disease prevention and treatment continues to gain attention as a desired approach for preventing or delaying disease [91]. Both human and preclinical studies suggest synergism of polyphenols such as lignans with existing therapeutics and, therefore, represent possible candidates for chemoprevention or as combination treatments with standard therapies such as chemotherapy, radiotherapy, immunotherapy, and gene therapy [259]. The overall results seem promising, yet the clinical evidence remains inconclusive [326,477,659,660,661,662]. Adoption of dietary polyphenols, like flaxseed lignans, into a “broad-spectrum therapeutic approach” will require an interdisciplinary approach combining prospective cohort studies investigating lignan exposure [326,477,481] with mechanistic studies to confirm the health benefits of flaxseed lignan interventions [4,654,655,656,657,658]. 

## 10. Conclusions

Dietary polyphenols represent a diverse array of chemical subgroups with evidence of variable efficacy in mitigating cancer risk and progression. Despite epidemiological support of possible benefit, these compounds lack general acceptance as therapeutic modalities in cancer treatment. This likely relates to an incomplete understanding of their mechanisms of action as well as a general lack of understanding of their absorption and pharmacokinetic characteristics resulting too often in exposure levels inadequate to address the disease process. Hence, an important purpose of this article was to review the scientific evidence of the role of flaxseed lignans in chemoprevention and on the growth, survival, and progression of malignant cells. This review consolidates years of unsystematic research with the flaxseed lignans and identifies lignans as having multiple targets and modes of action within the cancer phenotype. These multitargeted effects are broadly grouped as modulation of cell signaling and metabolism, cell growth and differentiation, cell motility and cytoskeletal dynamics, cell cycle, angiogenesis, and apoptosis. Such effects might explain the limited epidemiological evidence of lignan benefit in cancer, but a systematic approach, which includes lignan preclinical studies with translational relevance as well as clinical trials utilizing therapeutically relevant doses, will be needed to clarify their role in cancer. As other pharmaceuticals (e.g., the statin drugs) undergo repurposing to cancer treatment, a systematic investigation of polyphenolics such as the lignans might also harness their potential benefits towards chemoprevention and enhancement of patient longevity and quality of life. 

## Figures and Tables

**Figure 1 pharmaceuticals-12-00068-f001:**
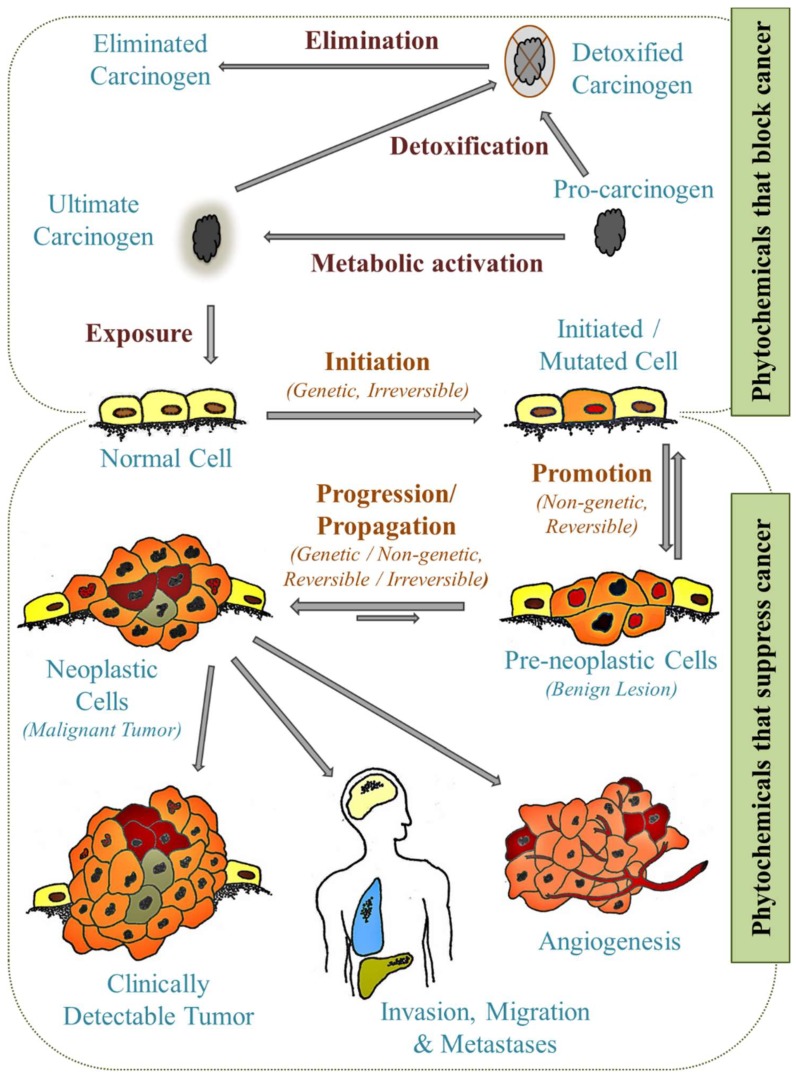
Polyphenolic phytochemicals (e.g., lignans) block and suppress carcinogenesis. Carcinogenesis is a multistage process of initiation, promotion, and progression. Carcinogens may initiate carcinogenesis by causing the conversion of a normal cell into an “initiated cell”, a process that is irreversible and involves genetic mutations. Initiated cells further transform into pre-neoplastic cells during the stage of promotion, and subsequently progress into neoplastic cells. Polyphenolic phytochemicals are capable of interfering with cellular and molecular processors in various stages of carcinogenesis. Phytochemicals may block cancer initiation through inhibition of procarcinogen activation into electrophilic species and their subsequent interaction with DNA. Alternatively, phytochemicals can stimulate carcinogen detoxification and their subsequent elimination from the body. Phytochemicals may suppress cancer by interfering with cancer promotion (a reversible process that involves nongenetic changes) or by regulating cancer progression, a complex process that involves both genetic and nongenetic changes as well as cell survival. Some polyphenols can act as blocking agents; others act as both blocking and suppressing agents, and some function as suppressing agents to modulate autophagy, cell cycle, and differentiation, thus affecting cancer cell proliferation. Adapted from references [3,4,67].

**Figure 2 pharmaceuticals-12-00068-f002:**
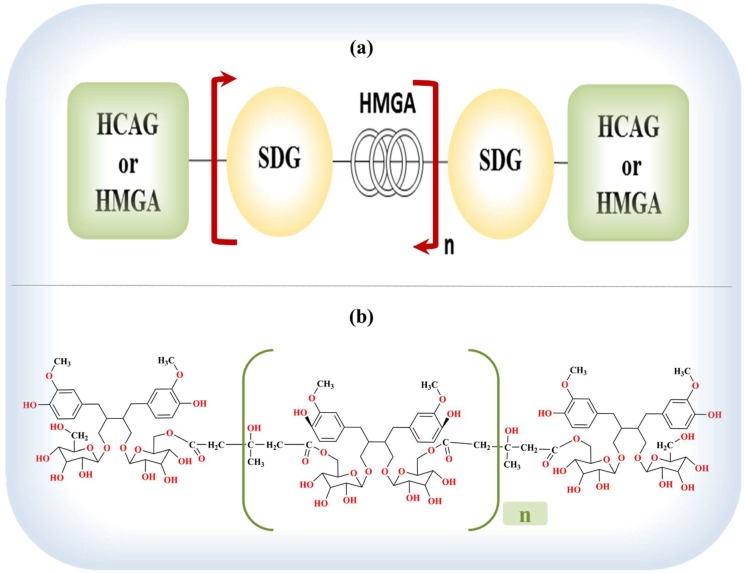
Chemical composition of flaxseed. (**a**) Schematic representation of the lignan macromolecule. The principal flaxseed lignan, secoisolariciresinol diglucoside (SDG), exists as a macromolecule in the flaxseed hull. This polymer complex is composed of five SDG structures held together by four hydroxy-methylglutaric acid (HMGA) residues with the hydroxycinnamic acids, p-coumaric glucoside (4-O-β-d-glucopyranosyl-p-coumaric acid or linocinnamarin) (CouAG), and ferulic acid glucoside (4-O-β-d-glucopyranosyl ferulic acid) (FeAG) as end units linked to the glucosyl moiety of SDG. The backbone moieties of this macromolecule are represented by the circles. The overlapping circles represent the linker molecule HMGA and the squares represent the terminal units. The terminal unit can be CouAG/FeAG or HMGA. (**b**) Postulated structure of the lignan oligomer. The SDG–HMGA polymer complex is converted into its monomer units—3-HMGA and SDG—by hydrolysis (average size, n = 3). Flaxseed contains high levels of the lignan oligomer (with ester linkages to HMGA, cinnamic acid, and other phenolic glucosides), which undergoes conversion to its aglycone, secoisolariciresinol (SECO), with further biotransformation into mammalian lignans by the action of the colonic bacteria in mammalian systems. Adopted from references [234,302,348,371,372].

**Figure 3 pharmaceuticals-12-00068-f003:**
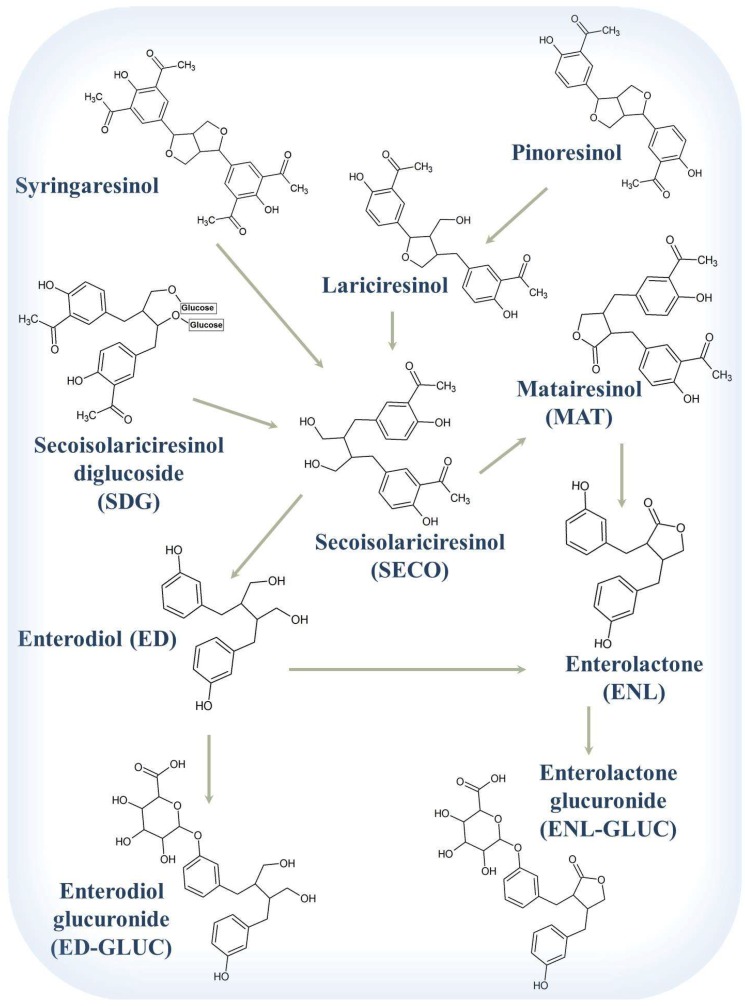
Lignan chemical structure and metabolites. Plant lignans are converted to various metabolites including the mammalian lignans (enterolactone (ENL) and enteroldiol (ED)) and their phase II metabolites such as glucuronide conjugates. The conversion of plant lignan secoisolariciresinol diglucoside (SDG) into the mammalian lignans can by separated into four catalytic reactions in order of O-deglycosylation (SDG to the its aglycone, SECO), O-demethylation (SECO to 2,3-bis (3’-hydroxybenzyl)butyrolactone/2,3-bis(3,4-dihydroxybenzyl)butane-1,4-diol), dehydrogenation (2,3-bis(3,4-dihydroxybenzyl)butane-1,4-diol to ED), and dihydroxylation (ED to ENL). Adopted from references [42,380,381,384].

**Figure 4 pharmaceuticals-12-00068-f004:**
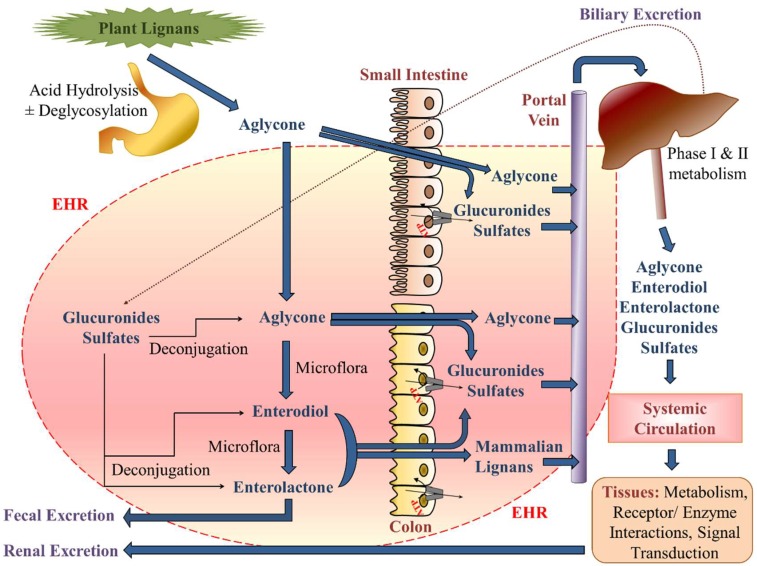
Flaxseed lignan absorption, first-pass metabolism, and enterohepatic recycling. The flaxseed lignan, secoisolariciresinol diglucoside (SDG), is biotransformed by bacteria in the gastrointestinal tract upon oral intake. Due to their lipophilicity, the aglycones and mammalian lignans may cross biological membranes via passive diffusion. With permeation into the enterocyte a portion of the aglycone and mammalian lignans undergo first-pass metabolism by phase II enzymes (e.g., UDP-glucuronosyltransferases (UGT), sulfotransferases (ST)). The polar, water-soluble glucuronide and sulfate conjugates require transport across the basolateral membrane of the intestinal epithelium by active transporters to gain access to the portal blood supply. Unmetabolized aglycone and mammalian lignans enter the hepatocyte by passive diffusion and undergo phase II metabolism by UGTs and STs. The conjugated metabolites are actively transported into the bile and can be reintroduced into the gastrointestinal tract. Here, they can be deconjugated and undergo reabsorption, a process called enterohepatic recirculation (EHR). The various lignans and their corresponding metabolites may elicit biological responses upon entering the systemic circulation by interacting with various enzymes, transporters, and other cell signaling macromolecules. Elimination of the conjugated metabolites can occur through either fecal or renal excretion. Adopted from reference [403].

**Figure 5 pharmaceuticals-12-00068-f005:**
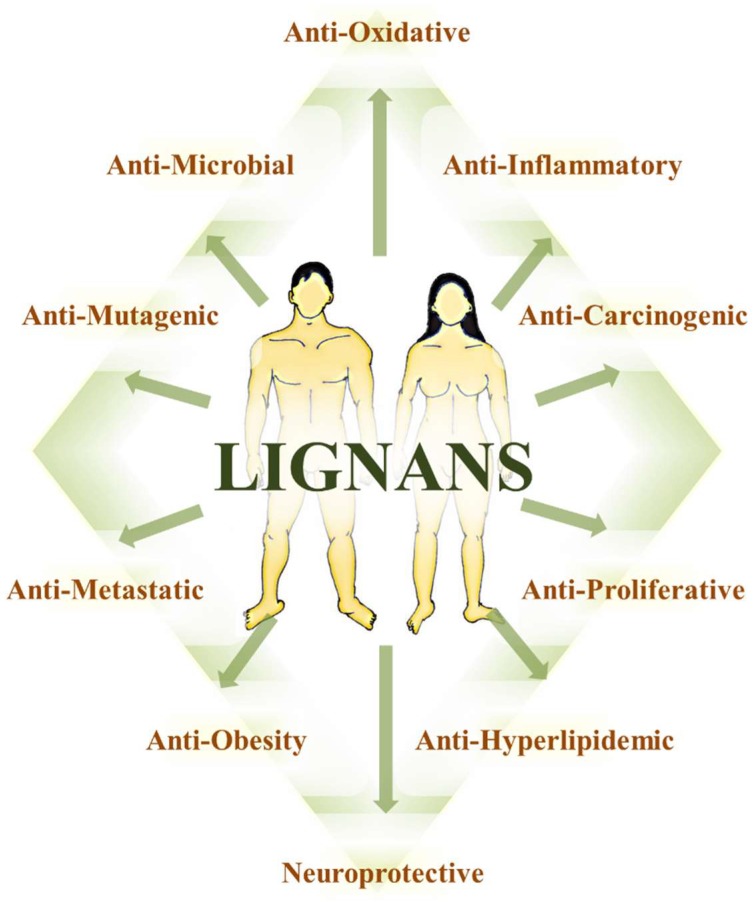
Protective health benefits of lignans. Lignans are polyphenolic phytochemicals that have varying biological activities under several contexts. Lignan containing diets or supplements can support general health as well as target many diseases.

**Figure 6 pharmaceuticals-12-00068-f006:**
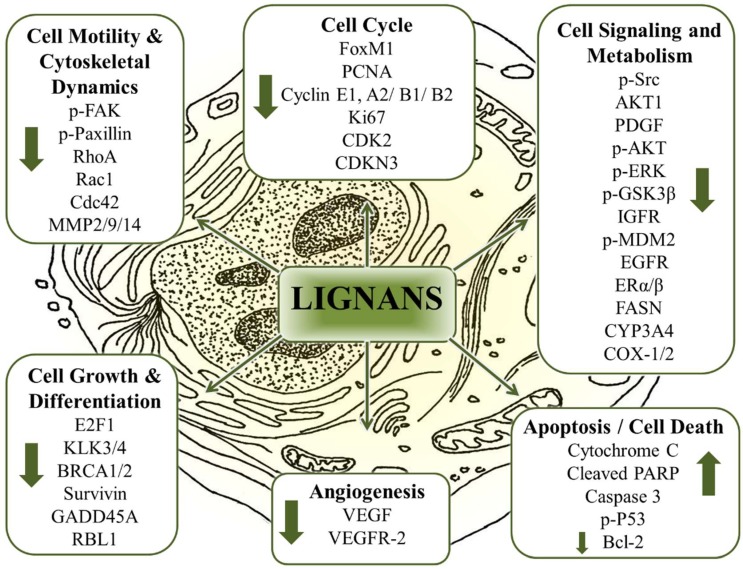
The cellular and molecular targets of lignans. Flaxseed lignans have the ability to target multiple pathways in cancer given the evidence from both *in vitro* and *in vivo* evaluations (Table 2). Cancer metastases can be inhibited by the modulation of the cytoskeleton and cell motility processors. Modulation of cell growth and differentiation as well as cell cycle arrest interfere with tumor proliferation and survival. Starving tumors by targeting angiogenesis as well as triggering apoptosis leads to inhibition of progression and survival. Interfering with different cell signaling pathways linking AKT and ERK modulates cell metabolism and disfavors progression and survival.

**Table 1 pharmaceuticals-12-00068-t001:** The classification of phytochemicals. Adopted from references [135,138,139].

Classification			Representative Members	Examples of Dietary Sources
Poly-Phenolics	Phenolic Acids	Hydroxycinnamic acids	p-Coumaric, caffeic, ferulic, sinapic	Barley, eucalyptus, coffee, Arabidopsis, Hibiscus, cereal grains
		Hydroxybenzoic acids	Gallic, vanillic, syringic, ellagic	Chestnuts (boiled or roasted), witch hazel, tea leaves, oak bark, rhubarb, pomegranate, grapes, chocolate, wine
	Lignans	Plant Lignans	sesamin, secoisolariciresinol diglucoside, lariciresinol, isolariciresinol, 7-hydroxymatairesinol, matairesinol, pinoresinol, arctigenin, syringaresinol, asarinin	Flaxseed, pumpkin, sunflower, poppy, rye, oats, barley, wheat, oat, rye, berries
		Mammalian Lignans (enterolignans)	Enterodiol, enterolactone	
	Stilbenes			Grapes
	Other Phenolics	Coumarins		Tonka bean, vanilla grass
		Tannins		Eucalyptus, geranium
	Flavonoids	Flavonols	Quercetin, kaempferol, myricetin	Aloe Vera, European elderberry, soy, St John’s wort, tomatoes, red onions
		Flavones	Apigenin, luteolin	Celery, parsley, chamomile tea, green peppers, thyme, oregano
		Flavanols (catechins)	Catechin, epicatechin, epigallocatechin gallate	White tea, green tea, persimmon, pomegranate, cocoa beans
		Flavanones	Eriodictyol, hesperetin	Citrus fruits, rose hip, mountain balm
		Anthocyanidins	Cyanidin, pelargonidin, malvidin	Grapes, berries, red cabbage, red onions, plums, kidney beans, geranium
		Isoflavonoids	Genistein, glycitein	Lupin, fava beans, soy, coffee
Alkaloids				Poppy, tomatoes, potatoes
Carotenoids	α-carotene, β-carotene, lutein, zeaxanthin, lycopene			Carrots, broccoli, spinach, zucchini
Organosulfur compounds	Isothiocyanates, indoles, allyl sulfur compounds			Cabbage, broccoli, spinach, garlic, onions

**Table 2 pharmaceuticals-12-00068-t002:** Cellular targets modulated by flaxseed lignan and lignan metabolites in cancer ^1^.

Experimental System and Lignan *	Targets: Molecules (Protein/Gene)	Block’s Model	Hanahan and Weinberg’s Model	Vogelstein et al., Model
MDA-MB231 (BC) ENL *	↓Ki67, ↓PCNA, ↓FoxM1, ↓Cyclin E1, ↓Cyclin A2, ↓Cyclin B1 ↓Cyclin B2 [627]	Proliferation, Immortality, Treatment resistance	Sustaining proliferative signaling, Evading growth suppressors	Cell survival
	↓pFAK, ↓pPaxillin [627] ↓ERK-1/2, ↓NF-κB, ↓MAPK-p38, ↓CD44 [647]	Proliferation, Metastasis, Cell-to-cell communication and Immortality	Activating invasion & metastasis, Sustaining proliferative signaling	Cell survival, Cell fate
	↓uPA, ↓MMP-2, ↓MMP-9, ↑PAI-1, ↑TIMP-1, ↑TIMP-2 [643] ↓N-cadherin, ↓vimentin, ↑E-cadherin, ↑occludin, ↓Snail [647]	Differentiation, Metastasis	Activating invasion & metastasis	Cell fate
XM (MDA-MB231) SDG *	↑LIV-1, ↑↓ ZIP2, ZnT-1 [648]	Proliferation	Sustaining proliferative signaling	Cell survival
MO (basal-like BC) SDG *	↓Proinflammatory markers (F4/80, CRP), ↓p-p65 [649]	Inflammation	Tumor promoting inflammation	Cell survival
MO (MCF7) (BC) ENL *	↓VEGF, ↑PIGF [331]	Proliferation, Treatment resistance, Angiogenesis	Inducing angiogenesis	Cell survival
OVX MO (MCF-7) SDG *	↓ERα, ↓ERβ, ↓EGFR, ↓pS2, ↓IGF-1R, ↓BCL2 [341]	Apoptosis, Proliferation, Glycemia	Sustaining proliferative signaling, Resisting cell death	Cell survival
	↓pMAPK [341]	Proliferation	Sustaining proliferative signaling	Cell survival
MCF7, MDA-MB231 ENL *	↓MMP2, ↓MMP9 ↓MMP14, ± MMP11 [642]	Differentiation, Metastasis	Activating invasion & metastasis	Cell fate
A549, H60 (Lung cancer) ENL *	↓pFAK, ↓pSrc, ↓pPaxillin [628]	Proliferation, Metastasis, Cell-to-cell communication	Activating invasion & metastasis, Sustaining proliferative signaling	Cell survival, Cell fate
	↓RhoA, ↓Rac1, ↓Cdc42 [628]	Metastasis, Cell-to-cell communication	Activating invasion & metastasis	Cell fate
	↑↓FAK, PDGF signaling (AKT1, CCND3). ↓RhoA, Rac1, Cdc42, ↑ITGA2 [628]	Metastasis, Differentiation, Proliferation, Cell-to-cell communication	Activating invasion & metastasis, Sustaining proliferative signaling	Cell survival, Cell fate
MG-63 (Osteosarcoma) ENL and ED *	Biphasic (↑↓) – osteonectin, collagen I [650]	Proliferation, Differentiation, Cell-to-cell communication	Activating invasion & metastasis	Cell fate
	↑ALP, ↑osteopontin, ↑osteocalcin [650]	Proliferation, Differentiation, Metastasis, Cell-to-cell communication	Activating invasion & metastasis	Cell fate
WPMY-1 (PS) ENL *	↑GPER, ↑p-ERK, ↑P53, ↑P21, ↓Cyclin D1 [651]	Proliferation, Immortality	Sustaining proliferative signaling	Cell survival
Rat prostate SDG *	↑GPER [651]	Proliferation, Immortality	Sustaining proliferative signaling	Cell survival
WPE1-NA22, WPE1-NB14, WPE1-NB11, WPE1-NB26 and LNCaP (PC) ENL *	↑↓DNA licensing genes (GMNN, CDT1, MCM2, MCM7) [593]	Proliferation, Immortality, Treatment resistance	Sustaining proliferative signaling	Cell survival, Genome maintenance
	↓miR-106b cluster (miR-106b, miR-93, miR-25),↑PTEN [593]	Proliferation, Angiogenesis	Sustaining proliferative signaling	Cell survival
LNCaP ENL *	↓BRCA1, ↓CDK2, ↓CDKN3, ↓E2F1, ↓KLK3, ↓KLK4, ↓PCNA, ↓PIAS1, ↓PRKCD, ↓PRKCH, ↓RASSF1, ↓TPM1, ↓SLC43A1 [335]	Proliferation, Immortality, Differentiation, Treatment resistance	Sustaining proliferative signaling, Replicative immortality, Evading growth suppressors	Cell survival, Genome maintenance Cell fate
	↓BIRC5, ↓BRCA1, ↓BRCA2, ↓CCNB1, ↓CCNB2, ↓CCNF, ↓CCNG1, ↓CCNH, ↓CDC2, ↓CDC20, ↓CDK2, ↓CDK4, ↓CDK5R1, ↓CDKN1B, ↓CDKN3, ↓CHEK1, CKS1B, ↓CKS2, ↓DDX11, ↓GADD45A, ↓KNTC1, ↓KPNA2, ↓MAD2L1, ↓MCM2, ↓MCM3, ↓MCM4, ↓MCM5, ↓MKI67, ↓MRE11A, ↓PCNA, ↓RBL1, ↓RPA3, ↓SKP2, ↑CCND2 [335]	Proliferation, Immortality, Treatment resistance, Stress chemistry	Sustaining proliferative signaling, Evading growth suppressors	Genome maintenance, Cell survival
LNCaP MAT *	↓pAKT [629]	Treatment resistance, Apoptosis, Proliferation, Glycemia	Sustaining proliferative signaling	Cell survival
	↓DR4 [629]	Apoptosis, Proliferation, Immortality	Resisting cell death	Cell survival, Cell fate
	↓TRAIL-DISC proteins (c-FLIP_L/S_, caspase-8) [629]	Apoptosis, Proliferation	Sustaining proliferative signaling, Resisting cell death	Cell survival, Cell fate
	↑TRAIL-induced BID cleavage [629]	Apoptosis, Proliferation	Resisting cell death	Cell survival
LNCaP ENL *	↑Cytochrome c release, ↑cleaved caspase-3, ↑PARP [634]	Apoptosis, Proliferation, Glycemia, Immortality, Oxidation	Deregulated cellular energetics, and Genome instability and mutation	Cell survival
	↓pAKT, ↓pGSK-3β, ↓pMDM2, ↑P53 [634]	Apoptosis, Immortality, Proliferation	Sustaining proliferative signaling, Evading growth suppressors, Enabling replicative immortality	Cell survival
	↑Caspase cell death [634]	Apoptosis	Resisting cell death	Cell survival
PC3 (PC) ENL *	↓pIGF-R(IGF-1), ↓pAKT, ↓p-p70S6K1, ↓pGSK3β, ↓pCyclinD1, ↓pERK ½ [618]	Proliferation, Glycemia, Immortality	Sustaining proliferative signaling, Activating invasion & metastasis	Cell survival, Cell fate
	↓IGF-1 signaling [618]	Proliferation, Glycemia	Sustaining proliferative signaling	Cell survival, Cell fate
	↓FASN [213]	Proliferation, Treatment resistance	Sustaining proliferative signaling, Deregulated cellular energetics	Cell survival
HUVEC (endothelial) ENL *	↓VEGFR-2 [331]	Proliferation, Angiogenesis	Inducing angiogenesis	Cell survival
Adipocytes ENL *	↓ROS - oxidative damage, ↓DNMTs, ↓HDACs, ↓MBD2 [334]	Proliferation, Oxidation, Inflammation, Stress chemistry, Immortality		Cell fate, Genome maintenance
Colonocytes-YAMC ENL and ED *	↓Cyclin D1, ↓Bcl-2 [586]	Proliferation, Immortality, Apoptosis	Sustaining proliferative signaling, Resisting cell death	Cell survival
Colo201 (COC) ENL *	↓Bcl-2, ↓PCNA, ↑cleaved caspase-3 [22]	Apoptosis, Proliferation	Resisting cell death	Cell survival
Apc-Min (intestinal) Diet (flaxseed) *	↓COX-1, COX-2 [652]	Proliferation, Immortality, Inflammation	Sustaining proliferative signaling, Tumor promoting inflammation	Cell survival
Hens Flaxseed supplement *	↓COX-2 [583]	Proliferation, Immortality, Inflammation	Tumor promoting inflammation	Cell survival
	↓Prostaglandin E2, ↓ERα, ↓CYP3A4, ↓CYP1B1, ↓16-OHE1, ↑CYP1A1, ↑2-OHE1 [583]	Proliferation, Inflammation, Treatment resistance, Stress chemistry	Tumor promoting inflammation	Cell survival
Hela (CC) ENL *	↓Viral oncogene E6 [588]	Proliferation	Evading growth suppressors	Cell survival
	↓Survivin [588]	Apoptosis, Proliferation	Resisting cell death, Sustaining proliferative signaling	Cell survival
	↑pHistone H2AX [588]	Apoptosis, Immortality, Proliferation	Resisting cell death	Cell survival, Cell fate
Hela ED *	↑Caspase 3 [588]	Apoptosis	Resisting cell death	Cell survival
CaSki (CC) ENL *	↓Viral oncogene E7 [588]	Proliferation	Evading growth suppressors	Cell survival
	↓Bcl-2 [588]	Apoptosis, Treatment resistance	Resisting cell death	Cell survival
Hela and CaSki ENL *	↑P53 [588]	Proliferation, Apoptosis	Evading growth suppressors	Cell survival, Genome maintenance
	↑Bax [588]	Apoptosis, Treatment resistance	Resisting cell death	Cell survival
	**Targets: Cellular Processors**			
SKBR3 and MDA-MB231 (BC) ENL *	↓Cell viability with anticancer agents [55,608]	Proliferation, Treatment resistance, Stress chemistry, Apoptosis	Resisting cell death, Sustaining proliferative signaling, Evading growth suppressors	Cell survival
MDA-MB231 ENL *	↑Cell cycle S phase, ↓cell viability [627]	Apoptosis, Immortality, Proliferation	Sustaining proliferative signaling, Evading growth suppressors	Cell survival, Genome maintenance, Cell fate
	↓Actin cytoskeleton organization [627,647] ↓Epithelial–mesenchymal transition [647]	Proliferation, Metastasis	Sustaining proliferative signaling, Activating invasion & metastasis	Cell survival, Cell fate
	↓Migration, invasion [627,642]	Metastasis	Activating invasion & metastasis	Cell fate
	↓Actin, filopodia, lamellipodia [642]	Proliferation, Metastasis	Sustaining proliferative signaling, Activating invasion & metastasis	Cell survival, Cell fate
	Anticancer/metastatic/ proliferative/migratory/clonogenic [643]	Metastasis	Activating invasion & metastasis	Cell fate
MCF7 and MDA-MB231 SDG and ASECO *	↓Growth [653]	Proliferation	Sustaining proliferative signaling	Cell survival
ER+ BC (XM) ENL and ED *	↓Angiogenesis [451]	Angiogenesis	Inducing angiogenesis	Cell survival
WPMY-1 ENL *	↓proliferation, arrested cell cycle (G_0_/G_1_) [651]	Proliferation	Sustaining proliferative signaling	Cell survival
Rat model (PH) SDG *	↓Prostate enlargement, # papillary projections, thickness of cell layers [651]	Proliferation	Sustaining proliferative signaling	Cell survival
WPE1-NA22, WPE1-NB14, WPE1-NB11, WPE1-NB26 and LNCaP ENL *	↓Metabolic activity,↑doubling time [593]	Proliferation, Stress chemistry, Oxidation	Sustaining proliferative signaling, Deregulated cellular energetics, Evading growth suppressors	Cell survival, Cell fate
	Modulated cell cycle [593]	Proliferation, Immortality	Evading growth suppressors, Sustaining proliferative signaling	Cell survival, Genome maintenance
	↑Apoptosis [593]	Immortality, Apoptosis	Sustaining proliferative signaling, Resisting cell death	Cell survival
LNCaP ENL *	↑Sub-G0 and S, ↓G0/G1, ↓G2/M cell cycle [335]	Proliferation, Immortality	Sustaining proliferative signaling, Evading growth suppressors	Cell survival, Cell fate
	↓Cell density, ↓metabolic activity, ↓PSA, ↑apoptosis [335]	Proliferation, Apoptosis	Sustaining proliferative signaling, Resisting cell death, Deregulated cellular energetics	Cell survival, Cell fate
	↑Apoptosis with anticancer agents [335]	Apoptosis	Resisting cell death	Cell survival
	↓Mitochondrial membrane potential [634]	Treatment resistance, Stress chemistry, Glycemia, Oxidation, Proliferation, Apoptosis	Deregulated cellular energetics	Cell survival
LNCaP MAT *	Death receptor sensitizer (sensitizes TRAIL-induced apoptosis) [629]	Proliferation, Apoptosis	Sustaining proliferative signaling, Evading growth suppressors, Resisting cell death	Cell survival, Cell fate
	↑TRAIL-induced mitochondrial depolarization [629]	Proliferation, Apoptosis	Resisting cell death, Deregulated cellular energetics	Cell survival
PC3 ENL *	↓IGF-1 induced proliferation, ↓cell cycle arrest (G0/G1) [618]	Proliferation	Sustaining proliferative signaling, Evading growth suppressors	Cell survival
	↓IGF-1 induced migration [618]	Metastasis	Activating invasion & metastasis	Cell survival, Cell fate
A549 and H60 ENL *	↓Migration, invasion [628]	Metastasis	Activating invasion & metastasis	Cell fate
	↓Density F-actin fibers [628]	Metastasis, Proliferation	Activating invasion & metastasis, Sustaining proliferative signaling	Cell survival, Cell fate
YAMC ENL and ED *	↓Cell growth, ↑ apoptosis [586]	Proliferation, Apoptosis	Resisting cell death, Sustaining proliferative signaling, Evading growth suppressors	Cell survival
MG-63 ENL and ED *	Biphasic (↓↑)- cell viability, ALP activity [650]	Proliferation	Sustaining proliferative signaling	Cell survival, Cell fate
Mouse model ENL *	↓Estradiol-induced endothelial cell infiltration [331]	Metastasis	Activating invasion & metastasis	Cell survival, Cell fate
Colo201 ENL *	↑Apoptosis (sub-G1 cells),↑cell viability [22]	Proliferation, Apoptosis	Sustaining proliferative signaling, Evading growth suppressors, Resisting cell death	Cell survival
CC cells ENL *	↑Cell death, ↓ metabolic activity in p53+ [588]	Immortality, Proliferation, Treatment resistance, Glycemia, Apoptosis	Evading growth suppressors, Resisting cell death, Deregulated cellular energetics	Cell survival, Genome maintenance
	↑Apoptosis (Hela) [588]	Apoptosis	Resisting cell death	Cell survival
CC cells ENL and ED *	↓Cell survival [588]	Immortality, Proliferation, Apoptosis	Sustaining proliferative signaling, Evading growth suppressors	Cell survival, Genome maintenance
TR C33-A (CC) ENL and ED *	↓Promoter activity (Episomal, HPV oncoproteins) [588]	Proliferation	Sustaining proliferative signaling, Evading growth suppressors	Cell survival
Hela ENL *	↑p53 activity [588]	Immortality, Proliferation	Evading growth suppressors	Cell survival, Genome maintenance
	No DNA-breaks (genotoxicity) [588]	Proliferation, Apoptosis	Resisting cell death, Evading growth suppressors	Cell survival
Hela/CaSki ENL *	↑Apoptosis (Caspase 9, Caspase 3) [588]	Proliferation, Apoptosis	Resisting cell death	Cell survival
CaSki ED *	↑Caspase 3 activity [588]	Proliferation, Apoptosis	Resisting cell death	Cell survival

^1^ Note: Processors may include anything other than an individual protein/gene target expression such as cell cycle, invasion, motility, metastases, cell viability, apoptosis, cytoskeletal dynamics, ATP levels, metabolic rates, oxygen consumption, target activity, etc. Each molecule or processor can be related to multiple pathways and hallmarks indicated in the models, and therefore what is listed are some selected examples. The different types of lignans are indicated with an asteric (*); e.g., Lignan*. Lower case (simple) “p” in certain instances denotes “phosphorylated” protein. Refer to abbreviations.

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
