# Peer review of "Flaxseed Lignans as Important Dietary Polyphenols for Cancer Prevention and Treatment: Chemistry, Pharmacokinetics, and Molecular Targets"

_pharmaceuticals, 2019, doi:10.3390/ph12020068_

Round 1

Reviewer 1 Report

Major points-

The scope of the article, the figures and Tables 1 and 2 are strong points of this review.  I gather that this article is aimed more at clinicians.  While the review makes some points, I am not sure it makes a convincing case for clinical trials.   I can make a recommendation for acceptance after some minor changes mentioned below, but I ask the authors whether they could/should beef up the arguments?  One point is to bring mention of the Halifax Project early in the review, to emphasize that some clinicians have signed on to the general premise of the value of a very broad target drug approach.   

A second issue is making a case for flaxseed lignans vs. flavonoids like quercetin   or epigallocatechin gallate.  What strengths do the lignans have that should give them any priority? While this is hinted at, I didn’t see any real case made for flaxseed lignans.

While I looked at the Block et al digest of the Halifax Project, when I look at the more general discussion of multi-target cancer therapy, the treatment strategies still seem to be more focused on single pathways (multiple targets within) than the very broad treatment  thesis presented here; PARP and the so-called synthetic  lethality, for example.    

I think there should be a little more emphasis by way of reorganization of the presentation on results of animal studies where SDG, SECO, ED and ENL are administered IP, IV, etc.  These results are otherwise scattered in the review to support a range of other observations.  In Table 2, ED and ENL seem to the major compounds under study and it would be good to isolate findings on these compounds.  While your study on ENL showed adverse effects in rats, others did not report such outcomes and the compound along with SGD, SECO and ED seem to relatively safe to use, IV or IP ( Penttinen et al 2007;   Damdimopoulou et al 2011;  Ali I  et al 2016; Power KA et al 2006;  Danbara N et al 2005;  Saarinen NM et al  2002).  I find this study approach more interesting given the poor bioavailability of flaxseed lignan metabolites and quirky dependence on subsets of the gut microbiome and other factors.   While diets of flaxseed components are of interest in prevention, the more direct routes are of interest in treatment.

Minor points-

Some general editing needed.

Paragraph of lines 49-66; USA regulations?

Table 1- minor alignment issue with “hydroxycinnamic/  /c”, 3rd line of table.

Line 150-please clarify “inverse” to what part of diet?

Line 186-“60%” based on what measure?

Line 296-should use ref 5 here as well for other bioactive components.

Line 530-Is there a better reference than 274?

Line 924-reference 18-link didn’t get me to the article and I had problems once in Pubmed.

The conclusion section is a bit anemic . 

Author Response

Response to Reviewer 1  

Dear Reviewer/ Editor,

 We appreciate your efforts in providing such thought-provoking comments and suggestions. We have tried our best to accommodate listed comments and suggestions below in-order to strengthen our review.

Thank you once again, for you valuable time and support!

Sincerely,

F De Silva

Major points-

General Response:

Please refer to the significantly modified manuscript with regards to the edits. Table S2 was also included.

·       Point 1: The scope of the article, the figures and Tables 1 and 2 are strong points of this review.  I gather that this article is aimed more at clinicians.  While the review makes some points, I am not sure it makes a convincing case for clinical trials.   I can make a recommendation for acceptance after some minor changes mentioned below, but I ask the authors whether they could/should beef up the arguments?  

Response:

Please refer to the modifications made throughout the manuscript, for the changes indicated as listed below. The introduction was revised to lay the foundation, for the focus on polyphenols (e.g. lignans). Section 3 (Cancer, the Unmet Medical Need) was modified to bring in the element of inflammation and anti-inflammatory effects of polyphenols in support of targeting the cancer hallmarks. Section 4 (Cancer Prevention) was modified to emphasize the capabilities and roles of polyphenols in different stages of prevention. In addition, changes were made including the addition of several sub-sections accordingly, to support a more convincing case.

·       Point 2: One point is to bring mention of the Halifax Project early in the review, to emphasize that some clinicians have signed on to the general premise of the value of a very broad target drug approach.   

Response:

Section 5 (Alternate Approaches to the Malignant Disease) was included to explain the Halifax Project, CAM, and to highlight the role of phytochemicals in a broad-spectrum therapeutic approach. Lines: 188-211.

·       Point 3: A second issue is making a case for flaxseed lignans vs. flavonoids like quercetin   or epigallocatechin gallate.  What strengths do the lignans have that should give them any priority? While this is hinted at, I didn’t see any real case made for flaxseed lignans.

Response:

Please refer to the following sections in the modified manuscript. Sections (Dietary Polyphenols as Principal Phytochemicals for Malignant Disease, General Properties of Polyphenols and Evidence on Health, Challenges Associated with Cancer Prevention and Dietary Polyphenols, Linking Benefits of Flaxseed with Cancer Associated Chronic Diseases, and General Nutridynamic Effects of Polyphenols). Lines: 235-293, 294-313, 419-456, 740-758, and 314-331.

·       Point 4: While I looked at the Block et al digest of the Halifax Project, when I look at the more general discussion of multi-target cancer therapy, the treatment strategies still seem to be more focused on single pathways (multiple targets within) than the very broad treatment  thesis presented here; PARP and the so-called synthetic  lethality, for example.    

Response:

Please refer to the following changes in the manuscript. We wish to highlight the lignan mechanisms of action that relate to the hallmarks of cancer, therefore have not allocated much content, and including specific details in this review with respect to the Halifax project. Additionally, our goal is to provide an alternative low cost option for prevention and treatment of cancer. As the primary author experienced socioeconomic issues that limit populations in low and middle income countries in getting better health outcomes, the authors propose the use of lignans within the broad scope of CAM. In addition, such countries already have practises of traditional or indigenous or alternative medicine practices. However, the authors emphasize the need to align such practises with the evidence based scientific approach and specifically on nutrikinetics and nutridynamics.

·       Point 5: I think there should be a little more emphasis by way of reorganization of the presentation on results of animal studies where SDG, SECO, ED and ENL are administered IP, IV, etc.  These results are otherwise scattered in the review to support a range of other observations.  In Table 2, ED and ENL seem to the major compounds under study and it would be good to isolate findings on these compounds.  While your study on ENL showed adverse effects in rats, others did not report such outcomes and the compound along with SGD, SECO and ED seem to relatively safe to use, IV or IP (Penttinen et al 2007;   Damdimopoulou et al 2011;  Ali I  et al 2016; Power KA et al 2006;  Danbara N et al 2005;  Saarinen NM et al  2002).  I find this study approach more interesting given the poor bioavailability of flaxseed lignan metabolites and quirky dependence on subsets of the gut microbiome and other factors.   While diets of flaxseed components are of interest in prevention, the more direct routes are of interest in treatment.

Response:

Please refer to the significantly modified manuscript. We wish to highlight the lignan mechanisms of action primarily, and we have published a review before indicating the above suggestions. Additionally, several other reviews are available in the literature which discusses the various studies in detail; some include tables which have grouped studies accordingly. Although plant lignans are administered (SDG), they are finally biotransformed into the mammalian lignans (ED and ENL). Since the mammalian lignans and their phase II metabolites represent the major circulating lignan forms after oral administration, we and others are particularly interested in the mammalian lignans.

However, clinical use of purified SECO, ED or ENL is not practical due to economical issues. The use of lignans in IV or IP routes would increase cost, training, safety concerns, and lack of interest.  In general, patients prefer non-invasive (e.g. oral route) instead of invasive routes (e.g. IV or IP). The sole purpose of table 2 is to list the lignan targets and lignans’ anti-cancer effects in targeting different hallmarks on cancer. As mentioned in the manuscript, the use of a lignan enriched product with a higher percentage of SDG (therapeutically relevant dose) seems sufficient based on clinical trials and, we have previously published protocols that can be applied in designing clinical trials. In addition, a lignan enriched product can be conveniently used by patients and/or general public at home and/or office for treatment and/or prevention. Therefore, we have not allocated much content in this review focusing on study designs.

Minor points-

Some general editing needed.

·       Point 6: Paragraph of lines 49-66; USA regulations?

Response:

We did not want to focus too much on the regulatory aspects. Therefore, we kept that section short. However, we did add a couple statements to address this comment. Lines: 66-71, Section 2 (Growing Use of Naturally Derived Products).

·       Point 7: Table 1- minor alignment issue with “hydroxycinnamic/  /c”, 3rd line of table.

Response:

Changes were made, as stated above.

·       Point 8: Line 150-please clarify “inverse” to what part of diet?

Response:

Sentence was modified to clarify the context. Lines: 236-238

·       Point 9: Line 186-“60%” based on what measure?

Response:

Sentence was modified to clarify the context. Lines: 272-275

·       Point 10: Line 296-should use ref 5 here as well for other bioactive components.

Response:

Sentence was modified to clarify the context. Reference #[5,319-321], Line: 469

·       Point 11: Line 530-Is there a better reference than 274?

Response:

The sentence which is referred to, in this comment (line 530) was removed. Sentence and paragraph was modified to clarify the context. Lines: 717-733

·       Point 12: Line 924-reference 18-link didn’t get me to the article and I had problems once in Pubmed.

Response:

Correct link was added. Reference #48, Line: 56, 1022, and 1242

·       Point 13: The conclusion section is a bit anemic. 

Response:

Section 9 (Final Remarks), a new section at the end was added. Section 10 (Conclusions) has been edited to address this suggestion as well. Lines: 1014-1051

Submission Date

22 March 2019

Date of this review

02 Apr 2019 17:42:35

Reviewer 2 Report

p.p1 {margin: 0.0px 0.0px 0.0px 0.0px; font: 12.0px 'Helvetica Neue'} li.li1 {margin: 0.0px 0.0px 0.0px 0.0px; font: 12.0px 'Helvetica Neue'} ul.ul1 {list-style-type: hyphen}

Comments:

authors should discuss the results of the following study: PMID: 27943649. The report summarised epidemiological evidence regarding dietary polyphenols and cancer risk. Authors did not find strong evidence on the protective role of lignan toward cancer risk. However, a weak association between dietary lignan intake and breast cancer was noted. (Page 15, line 542); please underline the lack and need of further epidemiological studies in order to draw conclusions.

Authors should briefly describe clinical trials evaluating the effect of flax (seeds, flour, oil etc.) on oxidative stress and inflammatory markers as this mechanism are associated with risk of several cancers.

The paragraph “General Properties of Polyphenolic Phytochemicals” should include brief description of epidemiological and clinical evidence of polyphenols effects toward human health.

page 29, line 816 “…the epidemiological evidence of lignan benefit in various cancers..” - based on recent report (PMID: 27943649) there is solely low evidence of lignan benefit toward breast cancer - please rephrase

Conclusions any comments on clinical evidence?

“in cancer stems largely from epidemiological evidence positively associating flaxseed lignan consumption with reduced risk of cancer development [5-7].” References 5-7 are not referring to epidemiological evidence, epidemiological evidence regarding flaxseed lignan intake and reduced risk of cancer is none, as published studies did not provide subgroup analysis on the association between flaxseed lignan intake and cancer risk, please rephrase.

Author Response

Response to Reviewer 2  

Dear Reviewer/ Editor,

I greatly appreciate your efforts in providing valuable comments and great suggestions in order to improve this manuscript. Some of the suggestions were an eye-opener for us, which led to the inclusion of additional sub-sections to support the role of lignans in cancer prevention and treatment. We have tried to do our best to accommodate the below listed comments and suggestions.

Thank you very much, for your valuable time and support!

Sincerely,

F De Silva 

Comments and Suggestions for Authors

General Response:

Please refer to the significantly modified manuscript with regards to the edits. Table S2 was also included.

Comments:

·       Point 1: Authors should discuss the results of the following study: PMID: 27943649. The report summarised epidemiological evidence regarding dietary polyphenols and cancer risk. Authors did not find strong evidence on the protective role of lignan toward cancer risk. However, a weak association between dietary lignan intake and breast cancer was noted. (Page 15, line 542); please underline the lack and need of further epidemiological studies in order to draw conclusions.

Response:

Significantly modifications were made to this manuscript, in-order to address the suggestions and requests indicated in this comment. The listed PMID 27943649 was added. The authors accept the lack and need of further epidemiological studies, and therefore have indicated it, in several sections of the review as relevant, including the conclusion section. We have also provided explanations as to address the differences in clinical and preclinical studies as well as possible solutions to overcome such discrepancies. Lines: 704-733

·       Point 2: Authors should briefly describe clinical trials evaluating the effect of flax (seeds, flour, oil etc.) on oxidative stress and inflammatory markers as this mechanism are associated with risk of several cancers.

Response:

Please refer to the significantly modified manuscript, as it is indicated below. Sections 8 (Polyphenols of Flaxseed as Important Phytochemicals in Malignant Disease, Linking Benefits of Flaxseed with Cancer Associated Chronic Diseases) and Table S2. Lines: 458-489, and 740-758

·       Point 3: The paragraph “General Properties of Polyphenolic Phytochemicals” should include brief description of epidemiological and clinical evidence of polyphenols effects toward human health.

Response:

Modified manuscript includes the following to address this comment. Sections 6.2, and 7 (General Properties of Polyphenols and Evidence on Health, and Challenges Associated with Cancer Prevention and Dietary Polyphenols). Lines: 293-457

·       Point 4: Page 29, line 816 “…the epidemiological evidence of lignan benefit in various cancers..” - based on recent report (PMID: 27943649) there is solely low evidence of lignan benefit toward breast cancer - please rephrase

Response:

The edited manuscript has taken this comment into consideration and have rephrased as appropriate. Lines: 717-733, 433-456, and 1019-1051

·       Point 5: Conclusions any comments on clinical evidence?

Response:

We have added conclusions drawing from clinical evidence throughout the manuscript. As mentioned in the manuscript, the use of a lignan enriched product with a higher percentage of SDG (therapeutically relevant dose) seems sufficient based on clinical trials and, we have previously published protocols that can be applied in designing clinical trials. In addition, a lignan enriched product can be conveniently used by patients and/or general public at home and/or office for treatment and/or prevention. Additionally, Section 9 (Final Remarks), a new section at the end was added. Section 10 (Conclusions) has been edited to address this suggestion as well. Lines: 1014-1051

·       Point 6: “in cancer stems largely from epidemiological evidence positively associating flaxseed lignan consumption with reduced risk of cancer development [5-7].” References 5-7 are not referring to epidemiological evidence, epidemiological evidence regarding flaxseed lignan intake and reduced risk of cancer is none, as published studies did not provide subgroup analysis on the association between flaxseed lignan intake and cancer risk, please rephrase.

Response:

Please refer to the changes made in the introduction, as requested. Lines: 37-45

Submission Date

22 March 2019

Date of this review

28 Mar 2019 10:58:18

Round 2

Reviewer 1 Report

The corrections and additions are fine.  Check word spacing on line 689.

Reviewer 2 Report

Thank you for addressing the comments. No further revision is required.